# Limitation of Characterizing Implicit Regularization by Data-independent Functions

**Leyang Zhang**                                                          *leyangz2@illinois.edu*
*Department of Mathematics, College of Liberal Arts & Sciences*
*University of Illinois–Urbana Champaign*

**Zhi-Qin John Xu**                                                      *xuzhiqin@sjtu.edu.cn*
*School of Mathematical Sciences, Institute of Natural Sciences, MOE-LSC*
*Shanghai Jiao Tong University*

**Tao Luo**[*]                                                          *luotao41@sjtu.edu.cn*
*School of Mathematical Sciences, Institute of Natural Sciences, MOE-LSC*
*Shanghai Jiao Tong University*
*CMA-Shanghai, Shanghai Artificial Intelligence Laboratory*

**Yaoyu Zhang**[†]                                                      *zhyy.sjtu@sjtu.edu.cn*
*School of Mathematical Sciences, Institute of Natural Sciences, MOE-LSC*
*Shanghai Jiao Tong University*
*Shanghai Center for Brain Science and Brain-Inspired Technology*

**Reviewed on OpenReview:** *https://openreview.net/forum?id=140kSqm0uy*

## Abstract

In recent years, understanding the implicit regularization of neural networks (NNs) has become a central task in deep learning theory. However, implicit regularization is itself not completely defined and well understood. In this work, we attempt to mathematically define and study implicit regularization. Importantly, we explore the limitations of a common approach to characterizing implicit regularization using data-independent functions. We propose two dynamical mechanisms, i.e., Two-point and One-point Overlapping mechanisms, based on which we provide two recipes for producing classes of one-hidden-neuron NNs that provably cannot be fully characterized by a type of or all data-independent functions. Following the previous works, our results further emphasize the profound data dependency of implicit regularization in general, inspiring us to study in detail the data dependency of NN implicit regularization in the future.

## 1 Introduction

One of the greatest mysteries of neural networks (NNs) is their ability to generalize well without any explicit regularization even when they are heavily overparametrized (Breiman, 1995; Zhang et al., 2017). For conventional machine learning algorithms, without a regularization term, heavily overparameterized models easily overfit the data. However, for NNs, it has been empirically observed that, with proper initialization, their training trajectories are implicitly biased towards well-generalized solutions. Such a training-induced regularization effect is commonly referred to as implicit regularization and is a central issue for the deep learning theory.

Currently, our theoretical understanding of implicit regularization is very limited. To help us understand NNs better, we make a further step to explore the following basic theoretical questions about implicit

---

[*]Corresponding Author
[†]Corresponding Author

regularization: (i) How to define implicit regularization mathematically; (ii) What is the relation between implicit regularization and conventional explicit regularization; (iii) How to characterize implicit regularization. Questions (ii) and (iii) are closely related in the sense that if implicit and explicit regularization are equivalent, then we may expect to find an explicit regularization function to fully characterize any implicit regularization. In this work, we specifically address the relation between implicit regularization and a widely considered class of explicit regularization—regularization by a data-independent function. In our study, this problem is converted to whether there always exists a data-independent function $G$ over the parameter space whose value exactly quantifies the preference of a certain training process. For overparameterized linear models, specific nonlinear models and also NNs in the NTK regime, such a data-independent function $G$ can be exactly derived, detailly introduced in Section 2. On the other hand, it has been proved that, for specific problems like matrix factorization, stochastic convex optimization and one-neuron ReLU NN, implicit regularization cannot be explained by norms, strongly convex functions and data-independent functions (Razin & Cohen, 2020; Dauber et al., 2020; Vardi & Shamir, 2021).

In our work, we take a further step to propose two types of global nonlinear dynamical mechanisms beyond the description of various data-independent functions (see Section 5). Importantly, we provide two general recipes, i.e., Two-point and One-point Overlapping Recipes, for producing families of one-hidden-neuron NNs that realize these two dynamical mechanisms, respectively. We also prove that their implicit regularizations cannot be fully characterized by any data-independent functions. Based on these results, we believe such mechanisms commonly exist in the training dynamics of general NNs; in other words, the implicit regularization of NNs is in general data-dependent. Our contribution in this work is summarized as follows.

(a) We give a mathematical definition of regularization, and define implicit and explicit regularization accordingly (Section 4.1).

(b) We attempt to find the nature of implicit regularization, focusing on gradient descent. In particular, we propose two general dynamical mechanisms, i.e., Two-point and One-point Overlapping mechanisms, which put stringent constraints or even make it impossible to fully characterize implicit regularization by data-independent functions (Section 5).

(c) Following the two mechanisms, we present Two-point and One-point Overlapping Recipes. The examples they produce include rich classes of one-hidden-neuron NNs which realize one (or both) of these two mechanisms (Section 6). Then we show that One-point Overlapping Recipe can be extended to two-layer NNs with multiple neurons, meanwhile discuss the idea to generalize both recipes to multi-layer NNs and multi-sample loss functions.

(d) Specifically, we give examples concerning one-hidden-neuron NNs with Sigmoid and Softplus activations. Experiments on such examples are also used to support our results.

(e) Based on (Vardi & Shamir, 2021), we further emphasize the importance of data-dependence of implicit regularization in general, which should be carefully studied for NNs in the future.

## 2 Related Works

In recent years, many works have studied the implicit regularization (Kukačka et al., 2017) for various problems. Progress has been achieved for many of them, e.g., matrix/tensor factorization, deep linear neural networks, NNs in the NTK regime, linear and nonlinear models, and general nonlinear deep NNs. We recapitulate some of these works as follows.

For general non-linear NNs, empirical studies suggest that NNs have an implicit regularization towards low-complexity function during training process (Arpit et al., 2017; Kalimeris et al., 2019; Goldt et al., 2020; Jin et al., 2020). For example, the frequency principle (Xu et al., 2019; 2020; Rahaman et al., 2019; Zhang et al., 2021; Xu et al., 2022) quantifies the implicit regularization of "simple solution" by showing that NNs learn the data from low to high frequency, i.e., implicit low-frequency regularization. The deep frequency principle qualitatively explains why deep learning can be faster by empirically showing that the effective target function for a deeper hidden layer biases towards lower frequency during the trainin (Xu & Zhou,

2021). However, such low-complexity/low-frequency regularization of general deep non-linear models is hard to be characterized by an exact function in general. Only several special cases are studied, for example, the models linear w.r.t. trainable parameters, models linear w.r.t. both trainable parameters and inputs, and those with certain homogeneous properties.

Various studies have been done for the first kind of NNs, i.e., NNs that are linear w.r.t. trainable parameters but are non-linear w.r.t. the input. For example, NNs in the linear regime are studied by Luo et al. (2021) and the NTK regime is studied by Jacot et al. (2018). By considering functions in the phase domain, it has also been shown that gradient descent (GD) for the training of such NNs often picks a low-frequency function from multiple solutions (Zhang et al., 2021; Luo et al., 2020), and such behavior can be exactly formulated by a data-independent function. Another characterization of implicit regularization for NNs in the linear regime, presented in Zhang et al. (2020) and Mei et al. (2019), uses norm difference between the initial and learned parameters or between the initial and learned NN outputs. Finally, Chizat & Bach (2020) shows that infinitely wide two-layer neural networks in the linear regime with homogeneous activations can be fully characterized as a max-margin classifier in certain situations.

The study of the second kind of model, i.e., those linear w.r.t. both trainable parameters and inputs, yields a series of results as well. One of the focuses is deep linear NN. The implicit regularization due to depth in deep linear NNs are quantitatively studied and exploited; these include biasing towards simple functions to improve the generalization (Gissin et al., 2019) and accelerating the training by providing a regularization that can be approximated by a momentum with adaptive learning rates to accelerate the gradient descent (GD) (Arora et al., 2018). For others, Soudry et al. (2018) shows that GD takes the linearly fully-connected networks to solutions with implicit regularization of max-margin, while Gunasekar et al. (2018a) shows that GD takes linear convolutional networks to linear solutions with another penalty in the frequency domain. Besides, deep matrix factorization by deep linear networks with GD induces nuclear norm minimization of the learned matrix, leading to an implicit low-rank regularization (Gunasekar et al., 2018b; Arora et al., 2019; Chou et al., 2020).

As far as we know, only specific and limited models of the third kind, i.e., the homogeneous ones, have been studied. For example, Woodworth et al. (2020) studies simple homogeneous models for which the implicit bias of training with gradient descent can be exactly derived as a function of the scale of the initialization.

While there are fruitful progress in explicitly characterizing the implicit regularization of (at least partially) linear models, explicitly characterizing the implicit regularization in the training of general non-linear models is more new, and encounters much difficulty. Therefore, with a focus on NNs, another line of works considers constructing counter-examples that provably cannot be characterized explicitly by specific types of functions like norms, strongly convex functions or more general data-independent functions (Razin & Cohen, 2020; Dauber et al., 2020; Vardi & Shamir, 2021). We list some of them below.

Razin & Cohen (2020) proved that, under some conditions, the matrix completion task performed by a deep linear NN, when trained by gradient descent with mean square error, can converge to an infimum, but there is no minimum, that is, this infimum cannot be obtained. Thus, in this example the implicit bias of the deep linear NN can not be described by any norm. Another kind of example, given by Dauber et al. (2020), is based on stochastic convex optimization. More recently, Vardi & Shamir (2021) makes a step closer to general nonlinear NNs by providing examples of gradient flows for one-neuron ReLU NNs which converge to global minima. Based on zero-initialization and the manually-assigned derivative of ReLU at 0, they show that the training of such networks cannot be described by any useful data-independent functions, in other words, the training depends largely on data.

Compared to these previous attempts, our work, with a focus on (non-linear) NNs, makes a step further in characterizing the implicit biases. Importantly, we analyze the reason behind the failure in using data-independent functions to explicitly characterize implicit regularization in network training, presenting general mechanisms (Section 5.1 mechanism) and corresponding example construction recipes (Section 5.1). Our examples are all based on the recipes (see Section 5.2 and 5.3), which generate diverse and rich classes of one-hidden-layer NNs. These are more systematic and universal compared to the existing ones we know, such as NNs in NTK regime, or those employing a specific type of activation (e.g., ReLU). Third, we follow the usual set-up of NN training, always considering over-parametrized networks (the number of parameters

exceeds the number of samples). Therefore, due to the generality and close relation to application, our results highlight profound data-dependency of NN implicit regularization and provide a valuable insight for advancing the study in this area. Overall, our results emphasize the profound data-dependency of implicit regularization in NNs. This aspect warrants thorough exploration in future studies, given its relevance and potential impact on practical applications.

## 3 Preliminaries

We begin with definitions and notations we will use frequently throughout our discussion below. We start with activation functions and models.

**Definition 3.1.** $\sigma : \mathbb{R} \to \mathbb{R}$ *is a real-valued function which we call an activation function. Its reciprocal is denoted by $\tilde{\sigma}$ (provided that it exists), i.e., $\tilde{\sigma}(x) = \frac{1}{\sigma(x)}$ when $\sigma(x) \neq 0$.*

In this definition, no smoothness requirements are imposed on $\sigma$ (or $\tilde{\sigma}$), however, in our One-point Overlapping Recipe, we further require that $\sigma$ is continuously differentiable.

**Definition 3.2.** *A model is a parametrized function $g : \mathbb{R}^M \times \mathbb{R}^d \to \mathbb{R}$. For any $(\boldsymbol{\theta}, \boldsymbol{x}) \in \mathbb{R}^M \times \mathbb{R}^d$, $\boldsymbol{\theta}$ is the parameter of $g$ and $\boldsymbol{x}$ the input of $g$. Thus, for each $\boldsymbol{\theta} \in \mathbb{R}^M$ we have a function $g(\boldsymbol{\theta}, \cdot) : \mathbb{R}^d \to \mathbb{R}$ and the training of $g$ modifies $\boldsymbol{\theta}$.*

We will often consider a one-neuron network. In this case $g$ has the form $g(\boldsymbol{\theta}, \boldsymbol{x}) = a\sigma(\boldsymbol{w}^{\mathrm{T}}\boldsymbol{x})$, where $\boldsymbol{\theta} = (a, \boldsymbol{w}) \in \mathbb{R} \times \mathbb{R}^d$ is its parameter and we write $\boldsymbol{w}^{\mathrm{T}}\boldsymbol{x} = (\boldsymbol{w}, \boldsymbol{x})$ the inner product of $\boldsymbol{w}$ and $\boldsymbol{x}$ on $\mathbb{R}^d$.

Then we define our dataset and loss function for training a model.

**Definition 3.3.** *A dataset is denoted by $S = \{(\boldsymbol{x}_i, y_i) : i \in \mathcal{I}\} \in \mathbb{R}^d \times \mathbb{R}$ for a given index set $\mathcal{I}$. A loss function (with respect to a given dataset $S$) is denoted by $L_S = L(\cdot, S)$.*

An example of $L_S$ is

$$L_S(\boldsymbol{\theta}) = L(\boldsymbol{\theta}, S) = |a\sigma(\boldsymbol{\theta}^{\mathrm{T}}\boldsymbol{x}) - y|^2, \qquad S = (\boldsymbol{x}, y) \subseteq \mathbb{R}^d \times \mathbb{R},$$

where $\boldsymbol{\theta} = (a, \boldsymbol{w}) \in \mathbb{R}^{1+d}$. If $L_S$ has a minimum, we further denote the set of its global minima by $\mathcal{M}_S$. For example, if $\min L_S = 0$ then $\mathcal{M}_S = L_S^{-1}\{0\}$. In our discussion about implicit regularization below, we will focus on the scenarios (see 4.1) in which training dynamics converge to global minima. Such focus is common in the study of implicit regularization, for example, Vardi & Shamir (2021). In general $\mathcal{M}_S$ depends on $S$, and we shall see in the next few sections that the failure of characterizing implicit regularization by a data-independent function is closely related to the strong dependence of $\mathcal{M}_S$ on $S$.

Finally, we will write $\gamma$ for a parametrization of a curve as well as its image. More notations will be introduced in the later sections.

## 4 Regularization

In conventional machine learning problems, regularization is often realized by adding a specific term to the loss function, namely explicit regularization, to help solve most ill-posed problems. In contrast, one of the magics of NNs is that, as aforementioned, its training often finds a good solution, as if it does the regularization "implicitly" (Zhang et al., 2017). To make the future study of explicit and implicit regularization more systematic and unified, we revisit the notion of regularization in this section. Mathematical formulation of general regularization is provided, which goes beyond the scope of gradient flow (GF) or gradient descent (GD). Based on this, we define implicit regularization and explicit regularization accordingly. Finally, we consider implicit regularization of GF for a loss function and discuss two types of characterization of them, both involving data-independent functions (see Example (b) in Section 4.1). These characterizations will be our focus in the rest part of the paper.

### 4.1 Revisiting Regularization

We begin by defining the regularization in a general sense as a mapping between collections of algorithms. Let $g : \mathbb{R}^M \times \mathbb{R}^d \to \mathbb{R}$ be a model as before. We say $A$ is a *method* if it maps an arbitrary dataset $S$ to a subset $A(S)$ of $\mathbb{R}^M$. We call $A(S)$ the *solution set of A*.

**Definition 4.1** (Regularization). *Let $\mathcal{A}, \mathcal{A}'$ be two collections of methods that find solutions to the parameters of $g(\boldsymbol{\theta}, \cdot)$. A regularization (from $\mathcal{A}$ to $\mathcal{A}'$) is just any map $\mathcal{R} : \mathcal{A} \to \mathcal{A}'$, i.e., $\mathcal{R}$ assigns a method $A$ in $\mathcal{A}$ to some $\mathcal{R}(A) \in \mathcal{A}'$.*

The effect of this assignment is that $\mathcal{R}$ implicitly relates the solution set of $A$ to that of $\mathcal{R}(A)$, provided that both exist. Also note that while this definition emphasizes the mathematical essence of regularization in general, the mapping ($\mathcal{R}$) itself could be difficult to determine; instead, the study of it in practice may focus more on understanding the properties of such mappings between specific collections of methods. Examples of such are implicit and explicit regularizations.

For implicit regularizations, we focus on methods that find the global minima of a loss function $L_S$, for any given dataset $S$. Let $A_{\min}$ be one of such methods, namely

$$A_{\min}(S) = \{\boldsymbol{\theta}^* \in \mathbb{R}^M : L_S(\boldsymbol{\theta}^*) = \min_{\boldsymbol{\theta} \in \mathbb{R}^M} L_S(\boldsymbol{\theta})\} = \operatorname{argmin}_{\boldsymbol{\theta} \in \mathbb{R}^M} L_S(\boldsymbol{\theta}).$$

We also define $\mathcal{A}$ to be a collection of methods $A_{\min}$ finding the global minima of $L_S$. The implicit regularization (for GF) will then be a mapping associating each $A_{\min}$ to a gradient flow in $\mathbb{R}^M$. These notations will be used throughout the discussion below.

**Definition 4.2** (implicit regularization for GF). *Let $L$ be a loss function (Definition 3.3). Denote the gradient flow (GF) of $L_S$ starting at $\boldsymbol{\theta}_0$ by $A_{GF,\boldsymbol{\theta}_0}$, namely, $A_{GF,\boldsymbol{\theta}_0}$ is defined by*

$$\begin{cases} \dot{\boldsymbol{\theta}}(t) = -\nabla_{\boldsymbol{\theta}} L_S(\boldsymbol{\theta}); \\ \boldsymbol{\theta}(0) = \boldsymbol{\theta}_0. \end{cases} \tag{1}$$

*Then a regularization $\mathcal{R} : \mathcal{A} = \{A_{min} : A_{min} \text{ finds the global minima of } L_S\} \to \{A_{GF,\boldsymbol{\theta}_0} : \boldsymbol{\theta}_0 \in \mathbb{R}^M\}$ is called an implicit regularization of GF for $L$, or simply, an implicit regularization for $L$.*

For example, we can consider the gradient flows on the loss landscape of a linear model, i.e., $\{A_{\mathrm{GF},\boldsymbol{\theta}_0} : \boldsymbol{\theta}_0 \in \mathbb{R}^M\}$ is the collection of gradient flows with respect to the loss function

$$L(\boldsymbol{\theta}, S) = \sum_{i=1}^n |\boldsymbol{\theta} \cdot \boldsymbol{x}_i - y_i|^2, \quad \boldsymbol{\theta} \in \mathbb{R}^M, n < M.$$

For the sample $S = \{(\boldsymbol{x}_i, y_i)\}_{i=1}^n$ we require that $(\boldsymbol{x}_1, ..., \boldsymbol{x}_n)$ has full rank. Let $\mathcal{A} := \{A_{\boldsymbol{\theta}_0} : \boldsymbol{\theta}_0 \in \mathbb{R}^M\}$ where each $A_{\boldsymbol{\theta}_0}$ finds the point in $L^{-1}(0)$ which has the shortest distance to $\boldsymbol{\theta}_0$. Then we obtain a map $\mathcal{R} : \mathcal{A} \to \{A_{\mathrm{GF},\boldsymbol{\theta}_0} : \boldsymbol{\theta}_0 \in \mathbb{R}^M\}$ by $\mathcal{R}(A_{\theta_0}) = A_{\mathrm{GF},\boldsymbol{\theta}_0}$.

To motivate the study of implicit regularization, we then give the following definition of explicit regularization. We will also focus on methods that find global minima (but may not be those for $L$).

**Definition 4.3** (explicit regularization). *Let $L$ be the loss function as before. Given a collection $\mathcal{A}'$ of methods such that for any $A' \in \mathcal{A}'$, any given dataset $S$ and any $\boldsymbol{\theta}_0^* \in A'(S)$, we have*

$$J_S(\boldsymbol{\theta}_0^*, A') = \min_{\boldsymbol{\theta} \in \mathbb{R}^M} J_S(\boldsymbol{\theta}, A') \tag{2}$$

*for some function $J_S : \mathbb{R}^M \times \mathcal{A}' \to [-\infty, \infty]$ related to $L$. An explicit regularization for $L$ is a regularization (i.e., a map) $\mathcal{R}_{\mathcal{A}'} : \mathcal{A} \to \mathcal{A}'$. Here $\mathcal{A} := \{A_{min}\}$.*

**Examples**.

(a) Let $J_S(\boldsymbol{\theta}, A') = L(\boldsymbol{\theta}, S) + H(\boldsymbol{\theta}, A')$ for any given $H : \mathbb{R}^M \times \mathcal{A}' \to \mathbb{R}$. This is just the form of many commonly used explicit regularization in machine learning. For example, consider $H(\boldsymbol{\theta}, A') := \|\boldsymbol{\theta}\|_1$, $H(\boldsymbol{\theta}, A') = \|\boldsymbol{\theta}\|_2$, or more generally $H(\boldsymbol{\theta}, A') = \|\boldsymbol{\theta}\|_r$, $r \geq 1$.

(b) Consider as before $\mathcal{A}' = \{A_{\mathrm{GF},\boldsymbol{\theta}_0} : \boldsymbol{\theta}_0 \in \mathbb{R}^M\}$. Because each $A_{\mathrm{GF},\boldsymbol{\theta}_0}$ is just a GF in $\mathbb{R}^M$, it is determined by $\boldsymbol{\theta}_0$. This means we obtain a map $G : \mathbb{R}^M \times \mathbb{R}^M \to \mathbb{R}$ such that for any $\boldsymbol{\theta}_0^* \in A_{\mathrm{GF},\boldsymbol{\theta}_0}$,

$$G(\boldsymbol{\theta}_0^*, \boldsymbol{\theta}_0) = \min_{\boldsymbol{\theta} \in \mathcal{M}_S} G(\boldsymbol{\theta}, \boldsymbol{\theta}_0).$$

The construction of $G$ is possible in a trivial way: we may find some $c, c' \in \mathbb{R}$ with $c' > c$, then set $G(\boldsymbol{\theta}_0^*, \boldsymbol{\theta}_0) = c$ for any $\boldsymbol{\theta}_0 \in \mathbb{R}^M$ and any $\boldsymbol{\theta}_0^* \in A_{GF,\boldsymbol{\theta}_0}$, and $G(\boldsymbol{\theta}^*, \boldsymbol{\theta}) = c'$ otherwise. In certain situation, we can make $G$ behave much better. For example, if the gradient flows are on the loss landscape of a linear regression problem, we may simply set $G(\boldsymbol{\theta}^*, \boldsymbol{\theta}) = |\boldsymbol{\theta}^* - \boldsymbol{\theta}|$ for all $(\boldsymbol{\theta}^*, \boldsymbol{\theta}) \in \mathbb{R}^M \times \mathbb{R}^M$.

Also notice that neither $H$ nor $G$ depends on $S$. In such cases we will say $\mathcal{R}_{\mathcal{A}'}$ is *characterized by data-independent function $H$ (or $G$)*.

## 4.2 Characterization of Implicit Regularization

A direct approach to understand the implicit regularization $\mathcal{R}$ is to look at the value of certain data-independent function $G$ over $\mathcal{M}_S$ to determine the element chosen (or preferred) by $\mathcal{R}$. Depending on the amount of information about $\mathcal{R}$ provided by $G$, we classify the following two types of characterization of $\mathcal{R}$ by $G$.

**Definition 4.4.** *We say that an implicit regularization for $L$ is characterized by a data-independent function $G : \mathbb{R}^M \times \mathbb{R}^M \to \mathbb{R}$ if for any $S$ and any $\boldsymbol{\theta}_0 \in \mathbb{R}^M$, the operation*

$$\mathrm{argmin}_{\boldsymbol{\theta} \in \mathcal{M}_S} G(\boldsymbol{\theta}, \boldsymbol{\theta}_0) = \boldsymbol{\theta}_0^* \tag{3}$$

*is well defined, i.e., $\boldsymbol{\theta}_0^*$ exists and is unique. Here $\boldsymbol{\theta}_0, \boldsymbol{\theta}_0^*$ are the initial value and long-term limit of the GF for $L$, respectively.*

**Definition 4.5.** *We say that the implicit regularization for $L$ is characterized by a data-independent function $G : \mathbb{R}^M \times \mathbb{R}^M \to \mathbb{R}$ in the weak sense if for any $S$ and any $\boldsymbol{\theta}_0 \in \mathbb{R}^M$,*

$$\min_{\boldsymbol{\theta} \in \mathcal{M}_S} G(\boldsymbol{\theta}, \boldsymbol{\theta}_0) = G(\boldsymbol{\theta}_0^*, \boldsymbol{\theta}_0), \tag{4}$$

*where $\boldsymbol{\theta}_0, \boldsymbol{\theta}_0^*$ are the initial value and long-term limit of the GF for $L$, respectively.*

It is not difficult to see that if an implicit regularization is characterized by a data-independent function $G$, then $G$ characterizes it in the weak sense. In other words, Definition 4.4 is stronger than Definition 4.5. Moreover, note that a constant function $G$ on $\mathbb{R}^M \times \mathbb{R}^M$ characterizes any implicit regularization for $L$ in the weak sense. Thus, every implicit regularization for $L$ can be characterized in the weak sense, however, what are interesting are those non-trivial ones. Conversely, if for some implicit regularization $\mathcal{R}$, the only data-independent functions characterizing it in the weak sense are constant ones, then $\mathcal{R}$ cannot be characterized by data-independent function.

# 5 Overlapping Mechanisms and Examples

Let $\mathcal{R}$ be an implicit regularization of GF for a loss function $L$. By our definitions above, the study of $\mathcal{R}$ in essence is to trace the families of training trajectories of GF. In this section, we focus on the characterization of implicit regularization of GF for $L$ by a data-independent function $G$, proposing dynamical mechanisms that put stringent constraints on $G$ or even make data-independent characterization impossible. These are the Two-point Overlapping Mechanism (Lemma 5.1) and One-point Overlapping Mechanism (Lemma 5.2), both of which can be realized by one-hidden-neuron NNs with common activation functions. This will be shown by two numerical examples (in Section 5.2 and 5.3) using Sigmoid and Softplus, respectively. Furthermore, they serve as prototypes of our Two-point and One-point overlapping Recipes.

## 5.1 Overlapping Mechanisms

**Lemma 5.1** (Two-point Overlapping Mechanism). *Fix $\boldsymbol{\theta}_0 \in \mathbb{R}^M$. Let $I$ be an index set and $\{S_i\}_{i \in I}$ be a collection of datasets. For each $i \in I$, let $\boldsymbol{\theta}_i^* \in \mathcal{M}_{S_i}$ denote the long-term limit of the GF for $L(\cdot, S_i)$ starting*

at $\boldsymbol{\theta}_0$. Suppose that for any $i \in I$, there is some $j \in I\backslash\{i\}$ such that $\boldsymbol{\theta}_i^* \neq \boldsymbol{\theta}_j^*$ and $\{\boldsymbol{\theta}_i^*, \boldsymbol{\theta}_j^*\} \subseteq \mathcal{M}_{S_i} \cap \mathcal{M}_{S_j}$ (see Figure 1 for an example). Then the following results hold.

(a) The implicit regularization for $L$ cannot be characterized by any data-independent function $G : \mathbb{R}^M \times \mathbb{R}^M \to \mathbb{R}$.

(b) Any data-independent function $G$ that characterizes the implicit regularization for $L$ in the weak sense is constant on $\{\boldsymbol{\theta}_i^*\}_{i \in I}$.

(c) Any continuous data-independent function $G \in C(\mathbb{R}^M \times \mathbb{R}^M)$ that characterizes the implicit regularization for $L$ in the weak sense is constant on the closure of $\{\boldsymbol{\theta}_i^*\}_{i \in I}$.

*Proof.* See the proof of Lemma A.1 in Appendix. □

**Lemma 5.2** (One-point Overlapping Mechanism). *Fix $\boldsymbol{\theta}_0, \boldsymbol{\theta}_0^* \in \mathbb{R}^M$. Let $\{\gamma_i\}_{i=1}^M$ be $M$ trajectories of GF for $L$ from $\boldsymbol{\theta}_0$ to $\boldsymbol{\theta}_0^*$, such that $\lim_{t \to \infty} \frac{\dot{\gamma}_i(t)}{|\dot{\gamma}_i(t)|}$ exist for all $i$ and the limits are linearly independent. If the implicit regularization for $L$ is characterized by a data-independent function $G \in C^1(\mathbb{R}^M \times \mathbb{R}^M)$ in the weak sense, then $\nabla G(\cdot, \boldsymbol{\theta}_0)|_{\boldsymbol{\theta}_0^*} = 0$, where the derivative is taken with respect to the first entry of $G$. (see Figure 2 for an example)*

*Proof.* See the proof of Lemma A.2 in Appendix. □

The One-point Overlapping Mechanism puts stringent constraint on $G$. If this mechanism is further strengthened such that trajectories starting from $\boldsymbol{\theta}_0$ with different data $S$ can overlap at any point in a neighbourhood of $\boldsymbol{\theta}_0^*$, then the corresponding implicit regularization cannot be characterized by any data-independent function. This strengthened mechanism can be realized for special cases in experiment, and we will try to provide a general recipe for this mechanism in our future works.

Two-point Overlapping Mechanism (Lemma 5.1), which works for arbitrary function $G : \mathbb{R}^M \times \mathbb{R}^M \to \mathbb{R}$, is the heart of Two-point Overlapping Recipes. It will be used to prove Theorem 6.1. One-point Overlapping Mechanism (Lemma 5.2), on the other hand, is more specific in that it requires $G$ to be continuous. It is the heart of the One-point Overlapping Recipe and it will be used to prove Theorem 6.2.

In the following subsections, we provide concrete examples of one-hidden-neuron NNs with common activation functions that can realize each of the above mechanisms. These two specific examples further inspire our general recipes in Section 6 for producing rich classes of one-hidden-neuron NNs.

### 5.2 Example for Two-point Overlapping Mechanism

In this example, we consider the one-hidden-neuron NN with Sigmoid activation, i.e.,

$$f(\boldsymbol{\theta}, x) = f(w, a, x) = \frac{a}{1 + e^{-wx}}, \qquad \boldsymbol{\theta} = (w, a) \in \mathbb{R}^2.$$

and one-sample $\ell_2$ loss

$$L_S(\boldsymbol{\theta}) = L(\boldsymbol{\theta}, \{(x,y)\}) = |f(\boldsymbol{\theta}, x) - y|^2, \qquad S = \{(x,y)\} \in \mathbb{R}^2.$$

Notice that for any $S$, the global minimum of $L_S$ is 0 and $L_S^{-1}\{0\}$ is a curve in $\mathbb{R}^2$. Indeed, $f(\boldsymbol{\theta}, x) = y$ is equivalent to

$$a = y \cdot e^{-wx} + y,$$

so that $a$ is a function of $w \in \mathbb{R}$. Therefore, as illustrated in Figure 1, by properly choosing two singleton datasets $S_1 = (x_1, y_1)$ and $S_2 = (x_2, y_2)$, we may obtain two sets of global minima for $L(\cdot, S_1)$ and $L(\cdot, S_2)$, respectively, which intersect at two points. Then, assigning each of these two points as a long-time limit (for a trajectory of GF) denoted by $\boldsymbol{\theta}_1^*$ and $\boldsymbol{\theta}_2^*$ respectively, we "trace back" the trajectories to obtain two curves in the stable manifolds of $\mathcal{M}_{S_1}, \mathcal{M}_{S_2}$, respectively. Then we select a point $\boldsymbol{\theta}_0$ in their intersection. By this

procedure, we find a $\boldsymbol{\theta}_0$, two datasets $S_1$ and $S_2$ and two gradient trajectories $\gamma_1, \gamma_2$ converging to two points in $\mathcal{M}_{S_1} \cap \mathcal{M}_{S_2}$ as required by the Two-point Overlapping Mechanism (Figure 1) .

Thus, the implicit regularization for $L$ can only be characterized by a data-independent function $G : \mathbb{R}^2 \times \mathbb{R}^2 \to \mathbb{R}$ in the weak sense, because we must have $G(\boldsymbol{\theta}_1^*, \boldsymbol{\theta}_0) = G(\boldsymbol{\theta}_2^*, \boldsymbol{\theta}_0) = \min_{\boldsymbol{\theta} \in \mathcal{M}_{S_1}} G(\boldsymbol{\theta}, \boldsymbol{\theta}_0)$. Clearly, $\boldsymbol{\theta}_1^*$ and $\boldsymbol{\theta}_2^*$ cannot be differentiated without information from data by any data-independent function $G$. Therefore, as the GF trajectories differentiate $\boldsymbol{\theta}_1^*$ and $\boldsymbol{\theta}_2^*$, the corresponding implicit regularization must be data-dependent.

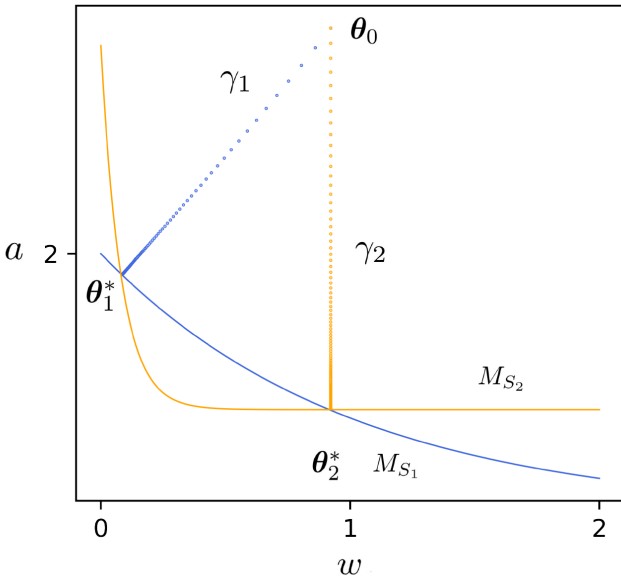

Figure 1: Two-point Overlapping Mechanism realized by a one-neuron-hidden NN with Sigmoid activation. Tracing back of gradient trajectories $\gamma_1, \gamma_2$ finds $\boldsymbol{\theta}_0$. This procedure inspires our Two-point Overlapping Recipe. We choose the initial point $\boldsymbol{\theta}_0 = (0.922, 2.868)$. The sample for (i) the blue lines is $(x_1, y_1) = (1, 1)$; (ii) the orange lines is $(x_2, y_2) = (12.307, 1.400)$.

### 5.3 Example for One-point Overlapping Mechanism

In this example, we consider another one-hidden-neuron NN with Softplus activation, i.e.,

$$f(\boldsymbol{\theta}, x) = f(w, a, x) = a \log(1 + e^{wx})$$

and the one-sample $\ell_2$ loss. Notice that if $y = -a\sigma(wx)$ then $f(w, -a, x) = y$, which means $(w, -a) \in L_S^{-1}\{0\}$ for $S = \{(x, -a\sigma(xw)\}$, for any $x \in \mathbb{R}$. Therefore, as illustrated in Figure 2, we first choose an initial point $\boldsymbol{\theta}_0 = (w_0, a_0)$. Then we use the one-element dataset $S = \{(x, -a_0\sigma(xw_0))\}$ with various $x$, by which we obtain distinct trajectories of GF from $\boldsymbol{\theta}_0$ to $\boldsymbol{\theta}^* = (w_0, -a_0)$, each one converging to $\boldsymbol{\theta}^*$ from different directions. In Figure 2, we show both the trajectories (dashed line) and $\mathcal{M}_S$'s (solid line), i.e., the sets of global minima of $L$, which clearly exhibits the One-point Overlapping Mechanism.

Thus, if the implicit regularization for $L$ is characterized by a data-independent function $G \in C^1(\mathbb{R}^2 \times \mathbb{R}^2) \to \mathbb{R}$ in the weak sense, then $\nabla G(\cdot, \boldsymbol{\theta}_0)|_{\boldsymbol{\theta}^*} = 0$, where the derivatives are taken with respect to the first entry of $G$.

## 6 Overlapping Recipes

In this section, we realize the overlapping mechanisms in Section 5 by providing two general recipes which produce rich classes of one-hidden-layer NNs, none of which can be (fully) characterized by a type of, or all data-independent functions. These recipes are exactly inspired by our numerical examples above; in fact, they can be viewed as generalizations of them.

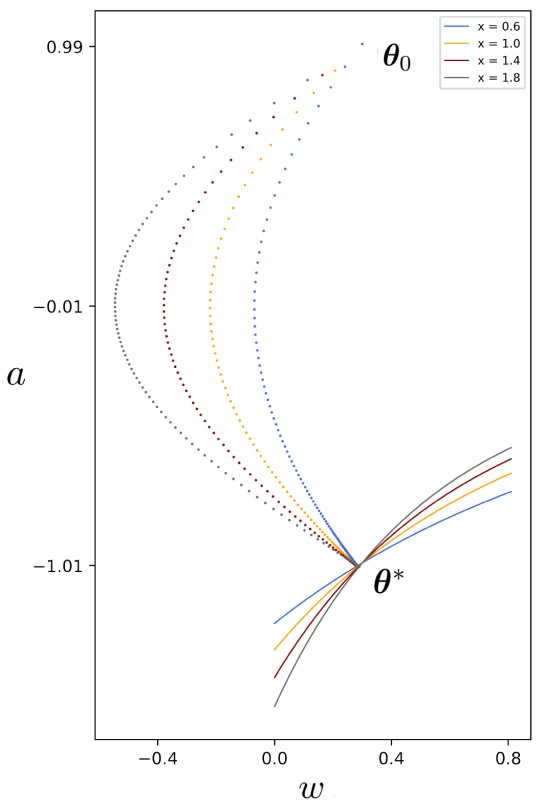

Figure 2: One-point Overlapping Mechanism realized by an one-neuron-hidden NN with Softplus activation. The dashed lines are gradient trajectories and the solid lines are $\mathcal{M}_S = L_S^{-1}\{0\}$ for different singleton datasets $S = \{(x, y)\}$. We choose initial value $\boldsymbol{\theta}_0 = (w_0, a_0) = (0.3, 1)$. Then $\boldsymbol{\theta}^* = (w_0, -a_0) = (0.3, -1)$. The dataset for (i) blue lines is $(x, y) = (0, 6, -a_0\sigma(0.6w_0))$; (ii) orange lines is $(x, y) = (1.0, -a_0\sigma(w_0))$; (iii) brown lines is $(x, y) = (1.4, -a_0\sigma(1.4w_0))$; (iv) grey lines is $(x, y) = (1.8, -a_0\sigma(1.8w_0))$.

### 6.1 Two-point Overlapping Recipe (Part A)

Our Two-point Overlapping Recipe produces one-hidden-neuron networks which realizes the Two-point Overlapping Mechanism (Lemma 5.1). It works by selecting a common initial value $\boldsymbol{\theta}_0$ for two gradient trajectories with respect to $S_1, S_2$, which converge to two points $\boldsymbol{\theta}_1^*, \boldsymbol{\theta}_2^* \in \mathcal{M}_{S_2} \cap \mathcal{M}_{S_1}$, respectively. In this procedure, the choice of $\boldsymbol{\theta}_0$ and $\boldsymbol{\theta}_1^*$ are (almost) arbitrary and one of the datasets ($S_1$ or $S_2$) can be chosen (almost) arbitrarily. Moreover, using this construction procedure, we can make $\sigma \in C^m(\mathbb{R})$ for any $m \in [0, \infty]$ and with other nice properties (monotonicity, periodicity, etc.).

The following procedure constructs $\sigma$ and finds $\boldsymbol{\theta}_0, \boldsymbol{\theta}_1^*, \boldsymbol{\theta}_2^*$, and $S_1, S_2$. For any $h \in \mathbb{R}^M$, define $P_h : \mathbb{R}^M \to \text{span}\{h\}$ be the orthogonal projection from $\mathbb{R}^M$ onto $\text{span}\{h\}$.

(a) Find $\boldsymbol{w}_0, a_0$ with $\boldsymbol{w}_0 \neq 0$. Find $\boldsymbol{\theta}_1^* = (\boldsymbol{w}_1^*, a_1^*)$ with $\boldsymbol{w}_1^* \neq \boldsymbol{w}_0$, $S_1 = (\boldsymbol{x}_1, y_1)$ with $\boldsymbol{x}_1 \neq 0$, and $\sigma_1 : \mathbb{R} \to \mathbb{R}$ such that the trajectory of GF for $L(\boldsymbol{\theta}, S_1) = |a\sigma_1(\boldsymbol{x}_1^{\mathrm{T}}\boldsymbol{w}) - y_1|^2$ starting at $\boldsymbol{\theta}_0$ converges to $\boldsymbol{\theta}_1^*$ as $t \to \infty$, and $L(\boldsymbol{\theta}_1^*, S_1) = 0$.

(b) Let $E_1 = \overline{\{\boldsymbol{x}_1^{\mathrm{T}}\boldsymbol{w}(t) \in \mathbb{R} : t \geq 0\}} \subseteq \mathbb{R}$.

(c) Find some $\boldsymbol{w}_2^*$ such that $\boldsymbol{x}_1^{\mathrm{T}}\boldsymbol{w}_2^* \notin E_1 \cup \{0\}$, and if $l$ is the line segment connecting $\boldsymbol{w}_0$ and $\boldsymbol{w}_2^*$, then $0 \neq P_h(\boldsymbol{w}_1^*) \notin P_h(l)$, where $h = \boldsymbol{w}_2^* - \boldsymbol{w}_0$.

(d) Find some $a_2^*$ and re-define $\sigma_1$ (if necessary) at $\{\boldsymbol{x}_1^{\mathrm{T}}\boldsymbol{w}_2^*\}$ such that $a_2^* a_0 \geq 0$ and $a_2^* \sigma_1(\boldsymbol{x}_1^{\mathrm{T}}\boldsymbol{w}_2^*) = y_1$. Let $\tilde{E}_1 = E_1 \cup \{\boldsymbol{x}_1^{\mathrm{T}}\boldsymbol{w}_2^*\}$.

(e) Find $\boldsymbol{x}_2 \in \text{span}\{\boldsymbol{w}_2^* - \boldsymbol{w}_0\}\setminus\{0\}$ with $\sup\{|\boldsymbol{x}_2|^{-1}|z| : z \in \tilde{E}_1\} < \min\{|P_h(\boldsymbol{w}_1^*)|, |P_h(\boldsymbol{w}_0)|, |P_h(\boldsymbol{w}_2^*)|\}$.

(f) Define $\sigma_2 : \mathbb{R} \to \mathbb{R}$ and $y_2$, such that i) $\sigma_2(\boldsymbol{x}_2^{\mathrm{T}}\boldsymbol{w}) = \sigma_1(\boldsymbol{x}_2^{\mathrm{T}}\boldsymbol{w})$ whenever $\boldsymbol{x}_2^{\mathrm{T}}\boldsymbol{w} \in \tilde{E}_1$, ii) $a_1^* \sigma_2(\boldsymbol{x}_2^{\mathrm{T}}\boldsymbol{w}_1^*) = y_2$, and iii) the trajectory $\gamma := (\gamma_{\boldsymbol{w}}, \gamma_a)$ of GF for $L(\cdot, S_2)$ starting at $\boldsymbol{\theta}_0$ converges to $\boldsymbol{\theta}_2^*$ as $t \to \infty$. Let $\sigma := \sigma_2$.

**Remark 6.1.** In Corollary A.1, we show that step (a) and (f) are well-established. This is achieved by Proposition A.1, which, based on the exponential function $e^x$, shows that given $S = \{(\boldsymbol{x}, y)\} \subseteq \mathbb{R}^d$ and two points $\boldsymbol{\theta}_0, \boldsymbol{\theta}^* \in \mathbb{R}^{d+1}$, there is a $\sigma : \mathbb{R} \to \mathbb{R}$ such that the GF of $L_S(a, \boldsymbol{w}) = |a\sigma(\boldsymbol{x}^T\boldsymbol{w}) - y|^2$ starting from $\boldsymbol{\theta}_0$ converges to $\boldsymbol{\theta}^*$. However, note that the choice of exponential function is just for the simplicity of proof; in general we could prove by using many other functions.

### 6.2 Two-point Overlapping Recipe (Part B)

The Two-point Overlapping Recipe (Part A) gives one-hidden-neuron networks that make it impossible to characterize the implicit regularization for $L$ by any data-independent function $G$. In fact, we can repeat the construction steps in Section 6.1 to obtain countably many datasets $\{S_n\}_{n=1}^{\infty}$ and countably many long-term limits of gradient trajectories $\{\boldsymbol{\theta}_n^*\}_{n=1}^{\infty}$ such that if the implicit regularization for $L$ is characterized by a data-independent function $G$ in the weak sense, then $G$ must be constant on $\{\boldsymbol{\theta} : \boldsymbol{\theta} = \boldsymbol{\theta}_n^*, n \in \mathbb{N}\}$. The detailed procedure is given below. As in Section 6.1, this procedure can also give a $\sigma$ of any degree of smoothness and with nice properties (monotonicity, periodicity, etc.).

The construction is described as follows. For $n = 1$, do the steps (a), (b) to obtain $\boldsymbol{\theta}_0, \boldsymbol{\theta}_1^*, \sigma_1$ and $E_1$. For $n \geq 2$, do the following steps.

(a) Find some $k \in \{1, ..., n-1\}$ and $\boldsymbol{w}_n^* \in \mathbb{R}^{M-1}$ such that $\boldsymbol{x}_k^{\mathrm{T}}\boldsymbol{w}_n^* \notin E_{n-1} \cup \{0\}$, and if $l$ is the line segment connecting $\boldsymbol{w}_0$ and $\boldsymbol{w}_n^*$ then $0 \neq P_h(\boldsymbol{w}_k^*) \notin P_h(l)$, where $h = \boldsymbol{w}_n^* - \boldsymbol{w}_0$.

(b) Find some $a_n^*$ and re-define $\sigma_{n-1}$ (if necessary) at $\{\boldsymbol{x}_k^{\mathrm{T}}\boldsymbol{w}_n^*\}$ such that $a_n^* a_0 \geq 0$ and $a_n^* \sigma_{n-1}(\boldsymbol{x}_k^{\mathrm{T}}\boldsymbol{w}_n^*) = y_k$. Let $\tilde{E}_{n-1} = E_{n-1} \cup \{\boldsymbol{x}_k^{\mathrm{T}}\boldsymbol{w}_n^*\}$.

(c) Find $\boldsymbol{x}_n \in \text{span}\{\boldsymbol{w}_n^* - \boldsymbol{w}_0\}\setminus\{0\}$ with $\sup\{|\boldsymbol{x}_n|^{-1}|z| : z \in \tilde{E}_{n-1}\} < \min\{|P_h(\boldsymbol{w}_k^*)|, |P_h(\boldsymbol{w}_0)|, |P_h(\boldsymbol{w}_n^*)|\}$.

(d) Define $\sigma_n : \mathbb{R} \to \mathbb{R}$ and $y_n$, such that i) $\sigma_n(\boldsymbol{x}_n^{\mathrm{T}}\boldsymbol{w}) = \sigma_{n-1}(\boldsymbol{x}_n^{\mathrm{T}}\boldsymbol{w})$ whenever $\boldsymbol{x}_n^{\mathrm{T}}\boldsymbol{w} \in \tilde{E}_{n-1}$, ii) $a_k^*\sigma_n(\boldsymbol{x}_n^{\mathrm{T}}\boldsymbol{w}_k^*) = y_n$, iii) the trajectory $\gamma := (\gamma_w, \gamma_a)$ of GF for $L(\cdot, S_n)$ starting at $\boldsymbol{\theta}_0$ converges to $\boldsymbol{\theta}_n^*$ as $t \to \infty$. Let $E_n := \tilde{E}_{n-1} \cup \boldsymbol{x}_n^{\mathrm{T}}\gamma_w$.

Finally, after doing this for countably many times, we have defined a function $\sigma_\infty$ on part of the real line. Now extend $\sigma_\infty$ to the whole real line. Let the extension be our activation function $\sigma$.

A simple induction argument shows that $G$ must be constant on $\{\boldsymbol{\theta} : \boldsymbol{\theta} = \boldsymbol{\theta}_n^*, n \in \mathbb{N}\}$. Indeed, suppose we have proved that

$$G(\boldsymbol{\theta}_1^*, \boldsymbol{\theta}_0) = G(\boldsymbol{\theta}_2^*, \boldsymbol{\theta}_0) = ... = G(\boldsymbol{\theta}_n^*, \boldsymbol{\theta}_0). \tag{5}$$

By our construction above, $\boldsymbol{\theta}_{n+1}^* \in \mathcal{M}_{S_k}$ for some $1 \le k \le n$, whence $G(\boldsymbol{\theta}_{n+1}^*, \boldsymbol{\theta}_0) \le G(\boldsymbol{\theta}_k^*, \boldsymbol{\theta}_0)$. Similarly, $\boldsymbol{\theta}_k^* \in \mathcal{M}_{S_{n+1}}$, whence $G(\boldsymbol{\theta}_k^*, \boldsymbol{\theta}_0) \le G(\boldsymbol{\theta}_{n+1}^*, \boldsymbol{\theta}_0)$. It follows that $G(\boldsymbol{\theta}_k^*, \boldsymbol{\theta}_0) = G(\boldsymbol{\theta}_{n+1}^*, \boldsymbol{\theta}_0)$, completing the induction step.

In Two-point Overlapping Recipe, we only find countably many points on which $G$ is constant. One may ask if we can find uncountably many such points. This is usually not true at least when $M = 2$ (so $d = 1$). In fact, for most $(w, a) \in \mathbb{R}^2$ and most $(x_0, y_0) \in \mathbb{R}^2$, there is a neighborhood $U$ of $(x_0, y_0)$ such that for $S_0 = \{(x_0, y_0)\}$, for any $S = \{(x, y)\} \subseteq U$, we cannot have

$$|L_S^{-1}\{0\} \cap L_{S_0}^{-1}\{0\}| \ge 2$$

and

$$(w, a) \in L_S^{-1}\{0\} \cap L_{S_0}^{-1}\{0\}$$

simultaneously ($|E|$ denotes the cardinality of a set $E$). Since each such $U$ contains a rational number and since $\mathbb{Q}$ is countable and dense in $\mathbb{R}$, it follows that we can find at most countably many points on which $G$ is constant. Moreover, this shows that the Two-point Overlapping Mechanism and the construction of our recipe above both utilizes the global property of the activation $\sigma$. A formal explanation of it is given in the following proposition. Recall that when $L((a, w), (x, y)) = 0$, $a = y/\sigma(xw) =: y\tilde{\sigma}(xw)$.

**Proposition 6.1** (Two-point Overlapping Recipe is global when $M = 2$). *Let $w \in \mathbb{R}$. Fix a point $(x_0, y_0) \in \mathbb{R}^2$ with $y_0 \ne 0$. Let $F : \mathbb{R}^2 \to \mathbb{R}$, $F(p, x) = \tilde{\sigma}(xw)\tilde{\sigma}(x_0p) - \tilde{\sigma}(xp)\tilde{\sigma}(x_0w)$. We have*

*(a) Suppose that $|F(p, x)| \ge C|p - w|^k|x - x_0|^r$ for some $C > 0$ and $r, k \in \mathbb{N}$ near $(w, x_0)$. Then for sufficiently small $\delta > 0$, if $0 < |x - x_0| < \delta$, $y \ne 0$ and $y\tilde{\sigma}(xw) = y_0\tilde{\sigma}(x_0w)$, there is no $p \in \mathbb{R}$ such that $0 < |p - w| < \delta$ and $y\tilde{\sigma}(xp) = y_0\tilde{\sigma}(x_0p)$.*

*(b) Suppose that $\tilde{\sigma} \in C^2$ and $\tilde{\sigma}(x_0w), \tilde{\sigma}'(x_0w) \ne 0$. Also suppose*

$$\frac{1}{w} - x_0 \left[ \frac{\tilde{\sigma}'(x_0w)}{\tilde{\sigma}(x_0w)} - \frac{\tilde{\sigma}''(x_0w)}{\tilde{\sigma}'(x_0w)} \right] \ne 0. \tag{6}$$

*Then for sufficiently small $\delta > 0$, if $0 < |x - x_0| < \delta$, $y \ne 0$ and $y\tilde{\sigma}(xw) = y_0\tilde{\sigma}(x_0w)$, there is no $p \in \mathbb{R}$ such that $0 < |p - w| < \delta$ and $y\tilde{\sigma}(xp) = y_0\tilde{\sigma}(x_0p)$. If, however, $DF(p, x_0) \equiv 0$ for $p$ near $w$ or $DF(w, x) \equiv 0$ for $x$ near $x_0$, then $\sigma$ is a power function near $x_0w$, i.e., $\sigma(x) = Cx^d$ for some $C, d \in \mathbb{R}$, when $x$ is sufficiently close to $x_0w$.*

*Proof.* See the proof of Proposition A.2 in Appendix. $\square$

**Remark 6.2.** We do not prove the case for $M > 2$, but we believe that this result also holds for $M > 2$. Namely, for most $(\boldsymbol{w}, a) \in \mathbb{R}^{d+1}$ and $S_0 = \{(\boldsymbol{x}_0, y_0)\} \subseteq \mathbb{R}^{d+1}$, there is a neighborhood $U$ of $(\boldsymbol{x}_0, y_0)$ such that for any $S = \{(\boldsymbol{x}, y)\} \subseteq U$, we cannot simultaneously have

$$|L_S^{-1}\{0\} \cap L_{S_0}^{-1}\{0\}| \ge 2$$

and

$$(\boldsymbol{w}, a) \in L_S^{-1}\{0\} \cap L_{S_0}^{-1}\{0\}.$$

Corollary 6.1 below indicates that Proposition 6.1 holds in general. Since we deal with neural networks, this corollary focuses on commonly-seen activation functions, including piecewise monomials, exponential activation, the Sigmoid activation and the Gaussian function.

**Corollary 6.1.** *Following the notations in Proposition 6.1, all the results below hold.*

(a) *Any $\sigma$ and $w, x_0 \neq 0$ such that $\sigma$ is a power function on a neighborhood of $x_0 w$ (this includes ReLU and PReLU and Heaviside) satisfies $F = 0$ near $(w, x_0)$.*

(b) *For any analytic activation $\sigma$ and any $x_0, w \in \mathbb{R}$ such that the zero locus of the function $F(p, x) = \tilde{\sigma}(xw)\tilde{\sigma}(x_0 p) - \tilde{\sigma}(xp)\tilde{\sigma}(x_0 w)$ satisfies*

$$F^{-1}\{0\} \cap U = \{(p, x) \in U : p = w\} \cup \{(p, x) \in U : x = x_0\}$$

*for some neighborhood $U \subseteq \mathbb{R}^2$ of $(w, x_0)$, we can find a sufficiently small $\delta > 0$ such that if $0 < |x - x_0| < \delta$, $y \neq 0$ and $y\tilde{\sigma}(xw) = y_0\tilde{\sigma}(x_0 w)$, there is no $p \in \mathbb{R}$ with $0 < |p - w| < \delta$ and $y\tilde{\sigma}(xp) = y_0\tilde{\sigma}(x_0 p)$.*

(c) *If $\sigma = e^x$ or $\sigma = e^{-x^2}$, then for any $x_0 \in \mathbb{R}$, we can find a sufficiently small $\delta > 0$ such that if $0 < |x - x_0| < \delta$, $y \neq 0$ and $y\tilde{\sigma}(xw) = y_0\tilde{\sigma}(x_0 w)$, there is no $p \in \mathbb{R}$ with $0 < |p - w| < \delta$ and $y\tilde{\sigma}(xp) = y_0\tilde{\sigma}(x_0 p)$.*

(d) *Let $w > 0$. If $\sigma = \frac{1}{1+e^{-x}}$, for any $x_0 \in (-\infty, w^{-1}) \cup (2w^{-1}, \infty)$, we can find a sufficiently small $\delta > 0$ such that if $0 < |x - x_0| < \delta$, $y \neq 0$ and $y\tilde{\sigma}(xw) = y_0\tilde{\sigma}(x_0 w)$, there is no $p \in \mathbb{R}$ with $0 < |p - w| < \delta$ and $y\tilde{\sigma}(xp) = y_0\tilde{\sigma}(x_0 p)$.*

*Proof.* See the proof of Corollary A.2 in Appendix. $\square$

### 6.3 One-point Overlapping Recipe

Clearly, Section 6.1 and 6.2 are not the only ways to negate the possibility that any one-hidden-neuron network can be characterized by a data-independent function $G$. We present another way below, called *One-point Overlapping Recipe*, which considers $C^1$ functions satisfying $G(\boldsymbol{p}, \boldsymbol{q}) = G(\boldsymbol{p} - \boldsymbol{q}, 0)$ for $\boldsymbol{p}, \boldsymbol{q} \in \mathbb{R}^M$; one such $G$ can be the Euclidean norm on $\mathbb{R}^M$. In this recipe, the choice of datasets are (almost) arbitrary, and we only require that $\sigma$ is differentiable, non-negative and strictly increasing on $\mathbb{R}$.

This recipe is described as follows.

(a) Find any $\sigma : \mathbb{R} \to \mathbb{R}^+$ such that $\sigma' > 0$.

(b) For each $n \in \mathbb{N}$, find any $\boldsymbol{\theta}_n = (\boldsymbol{w}_n, a_n)$ with $a_n \neq 0$. Select datasets $S_{n,k} = (\boldsymbol{x}_{n,k}, -a_n\sigma(\boldsymbol{x}_{n,k}^{\mathrm{T}}\boldsymbol{w}_n))$, such that the vectors

$$-\frac{\sigma(\boldsymbol{x}_{n,k}^{\mathrm{T}}\boldsymbol{w}_n)}{a_n\sigma'(\boldsymbol{x}_{n,k}^{\mathrm{T}}\boldsymbol{w}_n)} \left(\frac{1}{(x_{n,k})_1}, ..., \frac{1}{(x_{n,k})_d}\right), 1 \leq k \leq d$$

are linearly independent in $\mathbb{R}^d$.

(c) Repeat step (b) until we find enough $\boldsymbol{\theta}_n$'s with different values of $a_n$ ($\boldsymbol{w}_n$ can be arbitrary), as well as corresponding $S_{n,1}, ..., S_{n.d}$ for each $n \in \mathbb{N}$.

In (c), the word "enough" depends on the property of $G$ we would like to obtain. For example, in Lemma 6.2 we show that by carefully selecting one $\boldsymbol{\theta}_n$ and $d$ distinct datasets we can show that $\nabla G(\boldsymbol{p}, \boldsymbol{q}) = 0$ for some $\boldsymbol{p}, \boldsymbol{q} \in \mathbb{R}^M$; while in Theorem 6.2 we show that by carefully selecting countably many such points and datasets, we can show that $\nabla G(\boldsymbol{p}, \boldsymbol{q}) = 0$ on an affine subspace of $\mathbb{R}^M \times \mathbb{R}^M$.

The following two lemmas guarantee the validity of our One-point Overlapping Recipe.

**Lemma 6.1.** *Suppose that $\sigma > 0$ and $\sigma' > 0$ on $\mathbb{R}$. For any dataset $S = \{(x, y)\} \in \mathbb{R}\backslash\{0\} \times \mathbb{R}$ and any $\boldsymbol{\theta}_0 = (w_0, a_0)$, the trajectory of GF for $L(\cdot, S)$ has a long-term limit $\boldsymbol{\theta}_0^* \in \mathcal{M}_S$.*

**Remark 6.3.** This lemma is also used to construct concrete examples using the construction in Section 6.2. Moreover, the same result holds for $\sigma < 0$ and $\sigma' < 0$, because $L(\boldsymbol{\theta}) = |a\sigma(wx) - y|^2 = |a(-\sigma)(wx) - (-y)|^2$.

*Proof.* See the proof of Lemma A.3 in Appendix. $\qquad\square$

**Lemma 6.2.** *Let $\sigma : \mathbb{R} \to \mathbb{R}^+$ be differentiable and strictly increasing. Then*

> *(a) Suppose that $M = 2$. If the implicit regularization for $L$ is characterized by a data-independent function $G \in C^1$ in the weak sense, then there are some $\boldsymbol{\theta}_0, \boldsymbol{\theta}_0^* \in \mathbb{R}^2$ such that $\nabla G(\cdot, \boldsymbol{\theta}_0)|_{\boldsymbol{\theta}_0^*} = 0$, where the derivatives are taken with respect to the first entry of $G$.*

> *(b) The result in (a) also holds for general $M \geq 2$.*

*Proof.* See the proof of Lemma A.4 in Appendix. $\qquad\square$

## 6.4 Main Theorems

In this subsection, we summarize our examples based on the Two-point and One-point Overlapping recipes. Complete proof of the results are given in Appendix. Both theorems consider the following class of functions

$$\mathcal{G}_M = \{G \in C^1(\mathbb{R}^M \times \mathbb{R}^M : G(\boldsymbol{p}, \boldsymbol{q}) = G(\boldsymbol{p} - \boldsymbol{q}, 0)\}. \tag{7}$$

**Theorem 6.1.** *Based on the Two-point Overlapping Recipe, we have*

> *(a) For any $k \in \mathbb{N}$, we can construct an activation $\sigma \in C^k$ following Section 6.1, such that the implicit regularization for $L$ cannot be characterized by any data-independent function $G : \mathbb{R}^M \times \mathbb{R}^M \to \mathbb{R}$.*

> *(b) Following Section 6.2, for any $k \in \mathbb{N}$ we can find an activation $\sigma \in C^k$ such that if the implicit regularization for $L$ is characterized by a data-independent function $G \in C^1(\mathbb{R}^M \times \mathbb{R}^M)$ in the weak sense, then $G(\cdot, \boldsymbol{\theta}_0)$ is constant on an open set of $\mathbb{R}^M$ for some $\boldsymbol{\theta}_0 \in \mathbb{R}^M$.*

> *(c) Following Section 6.2, for any $k \in \mathbb{N}$ we can find an activation $\sigma \in C^k$ having the property that if the implicit regularization for $L$ is characterized by a data-independent function $G \in \mathcal{G}_M$ in the weak sense, then $G$ is constant.*

*Proof.* See the proofs of Theorem 6.1 (a), Theorem 6.1 (b) and Theorem 6.1 (c) in Appendix. $\qquad\square$

**Theorem 6.2.** *Let $\sigma : \mathbb{R} \to \mathbb{R}^+$ be differentiable and strictly increasing. Based on the One-point Overlapping Recipe, we have*

> *(a) $L$ cannot be characterized by any strongly convex data-independent function $G \in C^1(\mathbb{R}^M \times \mathbb{R}^M)$.*

> *(b) If the implicit regularization for $L$ is characterized by a data-independent function $G \in \mathcal{G}_M$ in the weak sense, then $G(\cdot, \boldsymbol{\theta})$ is constant on a line in $\mathbb{R}^M$ for any given $\boldsymbol{\theta}$.*

*Proof.* See the proof of Theorem 6.2 in Appendix. $\qquad\square$

**Corollary 6.2** (One-point Overlapping Recipe for Two-layer NNs)**.** *Fix $m, d \in \mathbb{N}$. Consider the two-layer neural network $g(\boldsymbol{\theta}, \boldsymbol{x}) = \sum_{k=1}^{m} a_k \sigma(\boldsymbol{w}_k \cdot \boldsymbol{x})$, where $\boldsymbol{\theta} = (\boldsymbol{w}_k, a_k)_{k=1}^m \in \mathbb{R}^{(d+1)m}$, and the corresponding loss function*

$$L(\boldsymbol{\theta}, (\boldsymbol{x}, y)) = |g(\boldsymbol{\theta}, \boldsymbol{x}) - y|^2 = \left|\sum_{k=1}^{m} a_k \sigma(\boldsymbol{w}_k \cdot \boldsymbol{x}) - y\right|^2.$$

*Suppose that $\sigma : \mathbb{R} \to \mathbb{R}^+$ is differentiable and strictly increasing. Based on One-point Overlapping Recipe, we have*

(a) *L cannot be characterized by any strongly convex data-independent function $G \in C^1(\mathbb{R}^{(d+1)m} \times \mathbb{R}^{(d+1)m})$.*

(b) *If the implicit regularization for $L$ is characterized by a data-independent function $G \in \mathcal{G}_M$ in the weak sense, then for any $\boldsymbol{\theta} \in \mathbb{R}^{(d+1)m}$ $G(\cdot, \boldsymbol{\theta})$ is constant on the set $\{(0, p_k)_{k=1}^m : p_k \in \mathbb{R}\}$.*

*Proof.* See the proof of Corollary 6.2 in Appendix. □

## 7 Conclusions and Discussion

### 7.1 Generalization of Overlapping Recipes

In this part we briefly discuss the generalization of our One-point Overlapping and Two-point Overlapping recipes (as well as corresponding mechanisms). We discuss the potential for our recipes to work for two-layer (fully-connected) NNs with multiple neurons with one-sample dataset, or even for more general models and loss functions.

Let's start with the Two-point Overlapping Recipe. Indeed, for this recipe, very few restrictions are put on the structure of the network or the loss functions; so in particular it can be generalized to a much larger set of models. To see this, consider a $\sigma$-dependent model $g = g_\sigma : \mathbb{R}^M \times \mathbb{R}^d \to \mathbb{R}$, and $L(\boldsymbol{\theta}, (\boldsymbol{x}, y)) = L_\sigma(\boldsymbol{\theta}, (\boldsymbol{x}, y)) = |g_\sigma(\boldsymbol{\theta}, \boldsymbol{x}) - y|^2$. The key of Two-point Overlapping Recipe is to "construct" the model $g$ by "constructing" $\sigma$, meanwhile taking the advantage that a convergent GF uses only partial information of $\sigma$. By looking at this recipe for one-neuron models (Section 6.1 and/or 6.2), to make the Two-point Overlapping Recipe work for $g_\sigma$ we basically need to

(a) Find $\boldsymbol{\theta}_1^*$, some dataset $S_1$ and some $\sigma_1$ such that the GF for $L_{\sigma_1}(\cdot, S_1)$ starting at $\boldsymbol{\theta}_0$ converges to $\boldsymbol{\theta}_1^*$, and $L_{\sigma_1}(\boldsymbol{\theta}_1^*, S_1) = 0$.

(b) Find $\boldsymbol{\theta}_2^*$ such that $g_{\sigma_1}(\boldsymbol{\theta}_2^*, S_1) = y$, namely, $L_{\sigma_1}(\boldsymbol{\theta}_2^*, S_1) = 0$.

(c) Find another dataset $S_2$ and $\sigma_2$ so that $g_{\sigma_2}(\boldsymbol{\theta}_2^*, S_2) = y$, and the GF for $L_{\sigma_2}(\cdot, S_2)$ converges to $\boldsymbol{\theta}_2^*$.

(d) Finally define $\sigma$ by appropriately "concatenating" $\sigma_1$ and $\sigma_2$.

Note that here we do not require that $S_1, S_2$ must be singletons. As long as the system is over-parametrized, these requirements are easy to satisfy, not only because they set few restrictions on the choice the activations, the samples, and the parameters we choose, but also because the requirements are loosely related to each other, e.g., requirement (b) does not have much to do with requirement (a).

The One-point Overlapping Recipe deals with the relationship between the partial derivatives of the loss function, so it naturally depends more on the structure of both the model and the loss function. We have shown that this recipe works for two-layer fully connected NNs as well. Unfortunately, currently we do not know how to generalize it to NNs with more layers, and/or to multi-sample loss functions. What we know is: to make it work we basically need to

(a) Find two distinct points point $\boldsymbol{\theta}_0, \boldsymbol{\theta}_0^* \in \mathbb{R}^M$ and some datasets $S_1, ..., S_M$.

(b) For each $1 \leq j \leq M$, the GF $\gamma_j$ for $L_{S_j}$ starting at $\boldsymbol{\theta}_0$ converges to $\boldsymbol{\theta}_0^*$.

(c) For each $1 \leq j \leq M$, $\gamma_j$ has a limiting direction, i.e., $\lim_{t \to \infty} \frac{\gamma_j(t)}{|\gamma_j(t)|}$ exists; moreover, these directions are linearly independent.

With such information we can conclude that $\nabla G(\cdot, \boldsymbol{\theta}_0)|_{\boldsymbol{\theta}_0^*} = 0$ as in Lemma 6.2.

## 7.2 Conclusion

In this work, we provide mathematical definitions of regularization, implicit regularization, and explicit regularization. We specify two levels of characterization of implicit regularization using a data-independent function $G$, i.e., (full) characterization and characterization in the weak sense. We delve into the nature of implicit regularization and address its challenges by proposing two general dynamical mechanisms, i.e., Two-point and One-point Overlapping mechanisms. These mechanisms make implicit regularization difficult to characterize or even impossible to characterize using data-independent functions. Additionally, we give numerical examples that realize these mechanisms with one-hidden-neuron NNs with Sigmoid or Softplus activations. These examples further inspire our development of Two-point and One-point Overlapping recipes that produce rich classes of one-hidden-neuron networks which realize these two mechanisms respectively. Last but not least, We show that our Two-point Overlapping Recipe depends on the global property of activation functions.

One strength of our work is that we proposed two mechanisms explaining why characterizing implicit regularization using data-independent functions often fails, and we have recipes that serve as general guidelines for the construction of examples. This systematic approach yields rich classes of common one-hidden-neuron NNs. Furthermore, as we have discussed before, our recipes and mechanisms have the potential to be extended to two-layer NNs with multiple neurons, or even more general models. In comparison, the existing examples mainly focus on more specific cases (e.g., specific set-up or specific kind of activations).The generality of our recipes thus suggests that it is generally difficult to characterize implicit regularization by data-independent functions, if not impossible.

On the other hand, our work does not fully explain the implicit regularization in NNs. For example, we do not know whether all the implicit regularization of NNs fall into one of our recipes and/or mechanisms. Neither are we clear about the practical implication of it. In particular, whether a data-dependent implicit regularization could help the generalization of DNNs remains an open problem for the future research.

While an implicit regularization is generally data-dependent, partial information about it may still be obtained by a data-independent function. Further studies should be conducted to mathematically determine details of such partial information. Besides, one may alternatively look for meaningful[1] data-dependent functions to characterize an implicit regularization. Since the non-equivalence between implicit and explicit regularization seem to depend on the global property of an activation function, one may also consider characterizing the training dynamics by a set of functions.

### Acknowledgments

This work is sponsored by the National Key R&D Program of China Grant No. 2022YFA1008200 (Z. X., T. L., Y. Z.), the National Natural Science Foundation of China Grant No. 12101402 (Y. Z.), No. 62002221 (Z. X.), No. 12101401 (T. L.), the Lingang Laboratory Grant No.LG-QS-202202-08 (Y. Z.), Shanghai Municipal of Science and Technology Project Grant No. 20JC1419500 (Y. Z.), Shanghai Municipal Science and Technology Key Project No. 22JC1401500 (T. L.), Shanghai Municipal of Science and Technology Major Project No. 2021SHZDZX0102.

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

## A  Appendix

**Lemma A.1** (Lemma 5.1). *Fix $\boldsymbol{\theta}_0 \in \mathbb{R}^M$. Let $I$ be an index set and $\{S_i\}_{i\in I}$ be a collection of datasets. For each $i \in I$, let $\boldsymbol{\theta}_i^* \in \mathcal{M}_{S_i}$ denote the long-term limit of the GF for $L(\cdot, S_i)$ starting at $\boldsymbol{\theta}_0$. Suppose that for any $i \in I$, there is some $j \in I \backslash \{i\}$ such that $\boldsymbol{\theta}_i^* \neq \boldsymbol{\theta}_j^*$ and $\{\boldsymbol{\theta}_i^*, \boldsymbol{\theta}_j^*\} \subseteq \mathcal{M}_{S_i} \cap \mathcal{M}_{S_j}$ (see Figure 1 for an example). Then the following results hold.*

*(a) The implicit regularization for $L$ cannot be characterized by any data-independent function $G : \mathbb{R}^M \times \mathbb{R}^M \to \mathbb{R}$.*

*(b) Any data-independent function $G$ that characterizes the implicit regularization for $L$ in the weak sense is constant on $\{\boldsymbol{\theta}_i^*\}_{i\in I}$.*

*(c) Any continuous data-independent function $G \in C(\mathbb{R}^M \times \mathbb{R}^M)$ that characterizes the implicit regularization for $L$ in the weak sense is constant on the closure of $\{\boldsymbol{\theta}_i^*\}_{i\in I}$.*

*Proof.*

(a) Suppose that the implicit regularization for $L$ is characterized by a data-independent function $G : \mathbb{R}^M \times \mathbb{R}^M \to \mathbb{R}$. Since $\boldsymbol{\theta}_i^* \in \mathcal{M}_{S_j} \backslash \{\boldsymbol{\theta}_j^*\}$ for some $j \in I$, we must have

$$G(\boldsymbol{\theta}_i^*, \boldsymbol{\theta}_0) > G(\boldsymbol{\theta}_j^*, \boldsymbol{\theta}_0). \tag{8}$$

Similarly, since $\boldsymbol{\theta}_j^* \in \mathcal{M}_{S_i} \backslash \{\boldsymbol{\theta}_i^*\}$, we must have

$$G(\boldsymbol{\theta}_j^*, \boldsymbol{\theta}_0) > G(\boldsymbol{\theta}_i^*, \boldsymbol{\theta}_0). \tag{9}$$

But then

$$G(\boldsymbol{\theta}_i^*, \boldsymbol{\theta}_0) < G(\boldsymbol{\theta}_j^*, \boldsymbol{\theta}_0) < G(\boldsymbol{\theta}_i^*, \boldsymbol{\theta}_0), \tag{10}$$

which is absurd.

(b) Argue in the same way as in (a), we can see that for any $i \in I$, there is some $j \neq i$ such that $G(\boldsymbol{\theta}_j^*, \boldsymbol{\theta}_0) = G(\boldsymbol{\theta}_i^*, \boldsymbol{\theta}_0)$. The desired result follows immediately.

(c) Clear from (b).

$\square$

**Lemma A.2** (Lemma 5.2). *Fix $\boldsymbol{\theta}_0, \boldsymbol{\theta}_0^* \in \mathbb{R}^M$. Let $\{\gamma_i\}_{i=1}^M$ be $M$ trajectories of GF for $L$ from $\boldsymbol{\theta}_0$ to $\boldsymbol{\theta}_0^*$, such that $\lim_{t\to\infty} \frac{\dot{\gamma}_i(t)}{|\dot{\gamma}_i(t)|}$ exist for all $i$ and the limits are linearly independent. If the implicit regularization for $L$ is characterized by a data-independent function $G \in C^1(\mathbb{R}^M \times \mathbb{R}^M)$ in the weak sense, then $\nabla G(\cdot, \boldsymbol{\theta}_0)|_{\boldsymbol{\theta}_0^*} = 0$, where the derivative is taken with respect to the first entry of $G$.*

*Proof.* Note that any $\gamma_i$ is eventually orthogonal to a null set of $L$ containing $\boldsymbol{\theta}^*$. The rest are clear. $\square$

**Proposition A.1.** *Let $\sigma(x) = e^x$ and $\boldsymbol{\theta}_0 = (\boldsymbol{w}_0, a_0) \in \mathbb{R}^M$ with $\boldsymbol{w}_0 \neq 0$. For any dataset $S$ the trajectory of GF for $L(\cdot, S)$ starting at $\boldsymbol{\theta}_0$ converges as $t \to \infty$. Conversely, for any $\boldsymbol{\theta}^* = (\boldsymbol{w}^*, a^*)$ such that $a^* \neq a_0$, there is a dataset $S$ such that the trajectory of GF for $L(\cdot, S)$ starting at $\boldsymbol{\theta}_0$ converges to $\boldsymbol{\theta}^*$ as $t \to \infty$.*

*Proof.* Fix any $S = (\boldsymbol{x}, y) \in \mathbb{R}^{M-1} \times \mathbb{R}$. We have $L(\boldsymbol{\theta}, S) = |ae^{\boldsymbol{x}^\mathrm{T}\boldsymbol{w}} - y|^2$. Thus,

$$\begin{cases} \dot{a} = -2(ae^{\boldsymbol{x}^\mathrm{T}\boldsymbol{w}} - y)e^{\boldsymbol{x}^\mathrm{T}\boldsymbol{w}} \\ \dot{w}_i = -2(ae^{\boldsymbol{x}^\mathrm{T}\boldsymbol{w}} - y)x_i ae^{\boldsymbol{x}^\mathrm{T}\boldsymbol{w}}, \end{cases} \tag{11}$$

Thus,

$$-2(ae^{\boldsymbol{x}^\mathrm{T}\boldsymbol{w}} - y)e^{\boldsymbol{x}^\mathrm{T}\boldsymbol{w}}x_i\dot{a}a = -2(ae^{\boldsymbol{x}^\mathrm{T}\boldsymbol{w}} - y)e^{\boldsymbol{x}^\mathrm{T}\boldsymbol{w}}\dot{\boldsymbol{w}}. \tag{12}$$

If $ae^{\boldsymbol{x}^{\mathrm{T}}\boldsymbol{w}} - y \neq 0$, we clearly have $x_i a\dot{a} = \dot{w}_i$. Otherwise, $\dot{a} = \dot{w}_i = 0$, so we still have $x_i a\dot{a} = \dot{w}_i$. Integrating on both sides of the equation, we see that

$$\int_0^t \dot{w}_i(u)du = x_i \int_0^t a(u)\dot{a}(u)du, \tag{13}$$

which yields $w_i(t) = w_i(0) + \frac{(a^2(t) - a_0^2)}{2}x_i$. Equivalently,

$$\boldsymbol{w}(t) = \boldsymbol{w}(0) + \frac{a^2(t) - a_0^2}{2}\boldsymbol{x}. \tag{14}$$

Thus, $\{(\boldsymbol{w}(t), a(t)) : t \in [0, \infty)\}$ is part of a parabola. This means as $t \to \infty$, either $\boldsymbol{w}(t), a(t)$ both diverge, or both of them converge. Suppose that $a^2(t) \to \infty$ as $t \to \infty$. Then there is some $N \in \mathbb{N}$ such that for any $t > N$ we have $\boldsymbol{x}^{\mathrm{T}}\boldsymbol{w}(t) > 0$. Whence $a(t)e^{\boldsymbol{x}^{\mathrm{T}}\boldsymbol{w}(t)} - y \to \infty$ as $t \to \infty$, which is a contradiction. It follows that $\lim_{t\to\infty} \boldsymbol{w}(t)$ and $\lim_{t\to\infty} a(t)$ exist.

Conversely, fix $\boldsymbol{\theta}^* = (\boldsymbol{w}^*, a^*)$ with $a^* \neq a_0$. Set

$$\boldsymbol{x} = \frac{2}{a^* - a_0}(\boldsymbol{w}^* - \boldsymbol{w}_0), \quad y = a^* e^{\boldsymbol{x}^{\mathrm{T}}\boldsymbol{w}^*}, \tag{15}$$

By our proof above, the trajectory of GF for $L(\cdot, (\boldsymbol{x}, y))$ has a long-term limit. Since $L(\boldsymbol{\theta}^*, (\boldsymbol{x}, y)) = 0$ and since $\boldsymbol{w}^* = \boldsymbol{w}(0) + \frac{a^{*2} - a_0^2}{2}\boldsymbol{x}$, this limit is $(\boldsymbol{w}^*, a^*) = \boldsymbol{\theta}^*$. □

**Corollary A.1.** *Based on Proposition A.1, we have the following results.*

(a) *There exist some $\boldsymbol{\theta}_0$, $\boldsymbol{\theta}_1^*$, $S_1$ and $\sigma_1$ such that step (a) in Two-point Overlapping Recipe (Part A) (Section 6.1) holds.*

(b) *There exist some $\sigma_2$ ($\sigma$) and $y_2$ such that step (f) in Two-point Overlapping Recipe (Part A) (Section 6.1) holds.*

*Proof.*

(a) Fix any $\boldsymbol{\theta}_0 = (\boldsymbol{w}_0, a_0)$ with $\boldsymbol{w}_0 \neq 0$ and any $\boldsymbol{\theta}_1^*$ with $\boldsymbol{w}_1^* \neq \boldsymbol{w}_0$ (this is the requirement of the recipe) and $a_0 \neq 0$ (this is the requirement of equation (15)). Find any $\boldsymbol{\theta}_1^* = (\boldsymbol{w}_1^*, a_1^*)$ such that (15) holds. The result follows immediately from Proposition A.1.

(b) Use equation (15) to find a dataset $\tilde{S}_2 = \{(\tilde{\boldsymbol{x}}_2, \tilde{y}_2)\}$ such that the trajectory of GF for $|ae^{\tilde{\boldsymbol{x}}_2^{\mathrm{T}}\boldsymbol{w}} - \tilde{y}_n|^2$ starting at $\boldsymbol{\theta}_0$ converges to $(\boldsymbol{w}_2^*, a_2^*)$ as $t \to \infty$. Note that we must have $\tilde{\boldsymbol{x}}_2 \in \text{span}\{\boldsymbol{w}_2^* - \boldsymbol{w}_0\}$. Also, since the sign of $a_2^*$ is the same as that of $a_0$, $\gamma_w$ is the line segment $l$ connecting $\boldsymbol{w}_0$ and $\boldsymbol{w}_2^*$. Therefore, we can set $y_2 = \tilde{y}_2$ and define $\sigma_2 : \mathbb{R} \to \mathbb{R}$ such that $\sigma_2(\boldsymbol{x}_2^{\mathrm{T}}\boldsymbol{w}) = \sigma_1(\boldsymbol{x}_2^{\mathrm{T}}\boldsymbol{w})$ whenever $\boldsymbol{x}_2^{\mathrm{T}}\boldsymbol{w} \in \tilde{E}_1$ and $\sigma_2(\boldsymbol{x}_2^{\mathrm{T}}\boldsymbol{w}) = e^{\tilde{\boldsymbol{x}}_2^{\mathrm{T}}\boldsymbol{w}}$ for $\boldsymbol{w} \in l$. Since $P_h(\boldsymbol{w}_1^*) \notin P_h(l)$, where $h = \boldsymbol{w}_2^* - \boldsymbol{w}_0$, it follows that we can re-define $\sigma_2$ at $\{\boldsymbol{x}_2^{\mathrm{T}}\boldsymbol{w}_1^*\}$ such that $a_1^*\sigma_2(\boldsymbol{x}_2^{\mathrm{T}}\boldsymbol{w}_1^*) = y_2$.

□

**Remark.** While our construction is based on the exponential activation function $\sigma(x) = e^x$, it can be based on any other activation function that satisfies: there are datasets $S_1, S_2$ such that the trajectories of GF for $L(\cdot, S_1), L(\cdot, S_2)$ starting at $\boldsymbol{\theta}_0$ converges to distinct $\boldsymbol{\theta}_1^*, \boldsymbol{\theta}_2^*$, respectively. For example, as we show in Section 5, the Sigmoid function is one candidate.

We now give the proof of our main theorems.

*Proof of Theorem 6.1 (a).* Suppose that the implicit regularization for $L$ is characterized by a data-independent function $G : \mathbb{R}^M \times \mathbb{R}^M \to \mathbb{R}$. The Two-point Overlapping Recipe (Part A) guarantees that $\boldsymbol{\theta}_1^* \in \mathcal{M}_{S_2} \backslash \{\boldsymbol{\theta}_2^*\}$ and $\boldsymbol{\theta}_2^* \in \mathcal{M}_{S_1} \backslash \{\boldsymbol{\theta}_1^*\}$. Applying Lemma 5.1 to the set

$$\{\boldsymbol{\theta}_1^*, \boldsymbol{\theta}_2^*\}, \tag{16}$$

we conclude that the implicit regularization for $L$ cannot be characterized by any data-independent function $G : \mathbb{R}^M \times \mathbb{R}^M \to \mathbb{R}$. It remains to show that $\sigma$ can be made as smooth as we want. To do this, let $\sigma_2$ be of $C^k$ when restricted to $\tilde{E}_1 \cup \boldsymbol{x}_2^{\mathrm{T}} \gamma_{\boldsymbol{w}}$. Extend $\sigma_2$ to a $C^k$ function on the whole $\mathbb{R}$. Since $\sigma = \sigma_2$ in Two-point Overlapping Recipe (Part A), $\sigma \in C^k$. $\qquad \square$

*Proof of Theorem 6.1 (b).* Follow the Two-point Overlapping Recipe (Part A) to obtain a $\sigma_1$, $\sigma_2$, $\boldsymbol{\theta}_0$, $\boldsymbol{\theta}_1^*$, $\boldsymbol{\theta}_2^*$, $S_1$ and $S_2$. For simplicity, we may further require that the construction is based on Proposition A.1 and corollary A.1, and $a_2^* \neq 0$, $a_2^* a_0 \geq 0$.

Find a small enough $r > 0$ such that for any $\boldsymbol{\theta}^* = (\boldsymbol{w}^*, a^*) \in B(\boldsymbol{\theta}_2^*, r)$, we have i) $\boldsymbol{\theta}_0 \neq \boldsymbol{\theta}^*$, ii) $0 \neq P_h(\boldsymbol{w}_1^*) \notin P_h(l)$, where $l$ is the line segment connecting $\boldsymbol{w}_0$ and $\boldsymbol{w}^*$ and $h = \boldsymbol{w}^* - \boldsymbol{w}_0$ and iii) $a^* \neq 0$. Geometrically and intuitively, $B(\boldsymbol{\theta}_2^*, r)$ is an open ball lying either above or below the $\boldsymbol{w}$-plane, and does not contain $\boldsymbol{\theta}_0$. Now find a countable, dense subset $\{\boldsymbol{\theta}_n^* = (\boldsymbol{w}_n^*, a_n^*)\}_{n=3}^{\infty}$ of $B(\boldsymbol{\theta}_2^*, r)$ such that for any distinct $i, j \geq 3$, $\boldsymbol{x}_1^{\mathrm{T}} \boldsymbol{w}_i^* \neq \boldsymbol{x}_1^{\mathrm{T}} \boldsymbol{w}_j^*$.

For $n \geq 3$, choose sufficiently large $\boldsymbol{x}_n$ such that step (c) in Two-point Overlapping Recipe (Part B) holds. Then use equation (15) to find a dataset $\tilde{S}_n = (\tilde{\boldsymbol{x}}_n, \tilde{y}_n)$ such that the trajectory $\gamma = (\gamma_{\boldsymbol{w}_n}, \gamma_{a_n})$ of GF for $|ae^{\tilde{\boldsymbol{x}}^{\mathrm{T}} \boldsymbol{w}} - y_n|^2$ starting at $\boldsymbol{\theta}_0$ converges to $\boldsymbol{\theta}_n^*$ as $t \to \infty$. Since the sign of $a_n^*$ equals $a_0$, $\gamma_{\boldsymbol{w}_n}$ is the line segment connecting $\boldsymbol{w}_0$ and $\boldsymbol{w}_n^*$. Set $y_n = \tilde{y}_n$. Define $\sigma_n : \mathbb{R} \to \mathbb{R}$ such that $\sigma_n(\boldsymbol{x}_n^{\mathrm{T}} \boldsymbol{w}) = \sigma_{n-1}(\boldsymbol{x}_n^{\mathrm{T}} \boldsymbol{w})$ whenever $\boldsymbol{x}_n^{\mathrm{T}} \boldsymbol{w} \in \tilde{E}_{n-1}$ and $\sigma_n(\boldsymbol{x}_n^{\mathrm{T}} \boldsymbol{w}) = e^{\tilde{\boldsymbol{x}}_n^{\mathrm{T}} \boldsymbol{w}}$ for $\boldsymbol{w} \in \gamma_{\boldsymbol{w}_n}$. Since $P_h(\boldsymbol{w}_1^*) \notin P_h(\gamma_{\boldsymbol{w}_n})$, where $h = \boldsymbol{w}_n^* - \boldsymbol{w}_0$, we can re-define $\sigma_n$ at $\{\boldsymbol{x}_n^{\mathrm{T}} \boldsymbol{w}_1^*\}$ such that $a_1^* \sigma_n(\boldsymbol{x}_n^{\mathrm{T}} \boldsymbol{w}_1^*)$.

Note that $\tilde{E}_{n-1}$ is the union of finitely many disjoint compact sets. Thus, after doing countably many times, we can obtain a $\sigma_{\infty}$ defined on a union of disjoint compact sets. Thus, by our proof A in Appendix, we can extend the $\sigma_{\infty}$ from this union of compact sets to be a $C^k$ function on $\mathbb{R}$. Since $\sigma = \sigma_{\infty}$ by our recipe, $\sigma \in C^k$.

Now our construction forces any data-independent $G : \mathbb{R}^M \times \mathbb{R}^M \to \mathbb{R}$ that characterizes the implicit regularization for $L$ in the weak sense to be constant on $\{\boldsymbol{\theta}_n^*\}_{n=2}^{\infty}$. Thus, if $G$ is continuous, it is constant on the closure of it, whose interior contains $B(\boldsymbol{\theta}_2^*, r)$. $\qquad \square$

**Remark.** Our choice of $e^{\boldsymbol{x}_n^{\mathrm{T}} \boldsymbol{w}}$ near $\boldsymbol{w}_0$ is not mandatory. Actually, we can let $\sigma(\boldsymbol{w}) = h(\boldsymbol{x}_n^{\mathrm{T}} \boldsymbol{w})$ for any $h : \mathbb{R} \to \mathbb{R}$ satisfying

(a) For any dataset $S$ the trajectory of GF for $L(\cdot, S)$ starting at $\boldsymbol{\theta}_0$ converges to $\boldsymbol{\theta}^*(S)$ as $t \to \infty$.

(b) The correspondence $S \mapsto \boldsymbol{\theta}^*(S)$ is a local continuous injection.

*Proof of Theorem 6.1 (c).* Do the construction in the proof of Theorem 6.1 (b) repeatedly, each time fixing some $\boldsymbol{\theta}_0$ and then finding $\boldsymbol{\theta}_0^*$ and $\{S_n\}_{n=1}^{\infty}$ carefully such that the closure of $\{\boldsymbol{\theta} : \boldsymbol{\theta} = \boldsymbol{\theta}_n^*, n \in \mathbb{N}\}$ is the translation of a "$2^M$-ant"[2] of $\mathbb{R}^M$ that does not contain $\boldsymbol{\theta}_0^*$. This shows that $G(\boldsymbol{\theta}, 0) = G(\boldsymbol{\theta} + \boldsymbol{\theta}_0, \boldsymbol{\theta}_0)$ for all $\boldsymbol{\theta}$ in some $2^n$-ant of $\mathbb{R}^M$. Choose different $\boldsymbol{\theta}_0$ and/or $\boldsymbol{\theta}_0^*$ to show that $G(\cdot, 0)$ must be constant on each $2^n$-ant of $\mathbb{R}^M$, whence $G(\cdot, 0)$ is constant. Since $G(\boldsymbol{p}, \boldsymbol{q}) = G(\boldsymbol{p} - \boldsymbol{q}, 0)$, $G$ must be constant on $\mathbb{R}^M \times \mathbb{R}^M$. $\qquad \square$

**Lemma A.3** (Lemma 6.1). *Suppose that $\sigma > 0$ and $\sigma' > 0$ on $\mathbb{R}$. For any sample $S = \{(x, y)\} \subseteq \mathbb{R} \times \mathbb{R}$ and any $\boldsymbol{\theta}_0 = (w_0, a_0)$, the trajectory of GF for $L(\cdot, S)$ starting at $\boldsymbol{\theta}_0$ has a long-term limit $\boldsymbol{\theta}_0^* \in \mathcal{M}_S$.*

---

[2] We define the $j$-th $2^M$-ant of $\mathbb{R}^M$ to be the closure of the set consisting of $\boldsymbol{\theta} \in \mathbb{R}^M$ such that the sign of the $i$-th component of $\boldsymbol{\theta}$ equals $2j_i - 1$, where $j = (j_{M-1}...j_0)_2$.

*Proof.* Note that the trajectory of the GF is characterized by

$$
\begin{cases}
\dot{a} = -2(a\sigma(xw) - y)\sigma(xw) \\
\dot{w} = -2(a\sigma(xw) - y)ax\sigma'(xw),
\end{cases}
\tag{17}
$$

with the initial value $w(0) = w_0$ and $a(0) = a_0$. Multiplying the two equations, we see that

$$
-2(a\sigma(xw) - y)x\sigma'(xw)a\dot{a} = -2(a\sigma(xw) - y)\sigma(xw)\dot{w}.
\tag{18}
$$

If $x = 0$, $L(\boldsymbol{\theta}, S) = |a\sigma(0) - y|^2$. In this case, the trajectory of GF for $L(\cdot, S)$ clearly converges. Now suppose that $x \neq 0$. If $a\sigma(xw) \neq y$, then

$$
a\dot{a} = \frac{\dot{w}\sigma(xw)}{x\sigma'(xw)}.
\tag{19}
$$

If $a(t)\sigma(xw(t)) = y$, then $\dot{a}(t) = \dot{w}(t) = 0$, so (19) also holds. Integrating on both sides of (19), we see that there is some strictly monotonic function $h$ (depends on $x$) such that

$$
\frac{1}{2}a^2(t) - \frac{1}{2}a_0^2 = h(w(t)) - h(w_0).
\tag{20}
$$

Thus, $w(t) = h^{-1}\left(h(w_0) + a^2(t)/2 - a_0^2/2\right)$ and thus the first equation in (17) becomes

$$
\dot{a} = -\left[a\sigma(xh^{-1}\left(h(w_0) + a^2(t)/2 - a_0^2/2\right)) - y\right]\sigma(xh^{-1}\left(h(w_0) + a^2(t)/2 - a_0^2/2\right)).
\tag{21}
$$

Define $\phi(s) = s\sigma(xh^{-1}(h(w_0) + s^2/2 - a_0^2/2)) - y$. If $z = h(w_0) - a_0^2/2$ then

$$
\begin{aligned}
\phi'(s) &= \sigma\left(xh^{-1}\left(\frac{s^2}{2} + z\right)\right) + s\sigma'\left(xh^{-1}\left(\frac{s^2}{2} + z\right)\right)x(h^{-1})'\left(\frac{s^2}{2} + z\right)s \\
&= \sigma\left(xh^{-1}\left(\frac{s^2}{2} + z\right)\right) + s^2\sigma'\left(xh^{-1}\left(\frac{s^2}{2} + z\right)\right)x(h^{-1})'\left(\frac{s^2}{2} + z\right).
\end{aligned}
\tag{22}
$$

Since $h$ is strictly increasing when $x > 0$ and strictly decreasing when $x < 0$, $x(h^{-1})'(s^2/2 + z)$ is always positive. Thus, $s^2\sigma'(xh^{-1}(z + s^2/2)x(h^{-1})'(z + s^2/2) > 0$. Moreover,

$$
\underline{\lim}_{s\to\pm\infty}\sigma\left(xh^{-1}\left(\frac{1}{2}s^2 + z\right)\right) > \sigma(xh^{-1}(z)),
\tag{23}
$$

where the right side of the inequality is positive. It follows $\phi'$ is bounded below and thus $\lim_{a\to\pm\infty}\phi(a) = \pm\infty$, so $\phi$ has a unique zero $a_0^*$. This is the point to which the $a$-component of the GF converges; moreover, if $w_0^* = h^{-1}(h(w_0) + a_0^{*2}/2 - a_0^2/2)$, then $(w_0^*, a_0^*)$ lies in $L^{-1}(\cdot, S)\{0\}$. This completes the proof. $\square$

**Lemma A.4** (Lemma 6.2). *Let $\sigma : \mathbb{R} \to \mathbb{R}^+$ be differentiable and strictly increasing. Then*

(a) *Suppose that $M = 2$. If the implicit regularization for $L$ is characterized by a data-independent function $G \in C^1$ in the weak sense, then there are some $\boldsymbol{\theta}_0, \boldsymbol{\theta}_0^* \in \mathbb{R}^2$ such that $\nabla G(\cdot, \boldsymbol{\theta}_0)|_{\boldsymbol{\theta}_0^*} = 0$, where the derivatives are taken with respect to the first entry of $G$.*

(b) *The result in (a) also holds for general $M \geq 2$.*

*Proof.*

(a) By Lemma 6.1, since $\sigma > 0$ and $\sigma' > 0$, for any dataset $S = \{(x, y)\} \subseteq \mathbb{R} \times \mathbb{R}$ with $x \neq 0$ and any $\boldsymbol{\theta}_0 = (w_0, a_0)$, the trajectory of GF for $L(\cdot, S)$ has a long-term limit $\boldsymbol{\theta}_0^* = (w_0^*, a_0^*) \in L^{-1}(\cdot, S)\{0\}$. Now, if $a_0 \neq 0$ and $S = (x, -a_0\sigma(xw_0))$, since

$$
L(\boldsymbol{\theta}_0, S) = |a_0\sigma(xw_0) - (-a_0\sigma(xw_0))|^2 = 4(a_0\sigma(xw_0))^2 > 0,
\tag{24}
$$

the continuity of $a\sigma(xw) - y$ ensures that $a_0^* \neq a_0$. Thus, $a_0^* = -a_0$. Then

$$h(w_0^*) = h(w_0) + a_0^{*2}/2 - a^2/2 = h(w_0) + 0. \tag{25}$$

Because $h$ is monotonic, we have $w_0^* = w_0$. Thus, $\boldsymbol{\theta}_0^* = (w_0, -a_0)$. Moreover,

$$\begin{aligned} \lim_{t\to\infty} \frac{\dot{a}(t)}{\dot{w}(t)} &= \lim_{\boldsymbol{\theta}\to\boldsymbol{\theta}_0^*} \frac{\sigma(xw)}{ax\sigma'(xw)} \\ &= -\frac{\sigma(xw_0)}{a_0 x \sigma'(xw_0)}. \end{aligned} \tag{26}$$

Suppose that $\lim_{t\to\infty} \frac{\dot{a}(t)}{\dot{w}(t)} = k$ for all $x \in \mathbb{R}$. Then since $\sigma' \neq 0$ on $\mathbb{R}$, $k \neq 0$ and thus $\frac{\sigma'(xw_0)}{\sigma(xw_0)} = \frac{1}{a_0 k}\frac{1}{x}$. Integrating both sides with respect to $x$, we can see that there are some non-zero constants $A, B \in \mathbb{R}$ such that

$$\log(A\sigma(xw_0)) = \frac{1}{a_0 k}\log(Bx). \tag{27}$$

Thus,

$$\sigma(xw_0) = \frac{(Bx)^{1/a_0 k}}{A}, \tag{28}$$

which implies that $\sigma$ is a monomial; but then $\sigma(0) = 0$, a contradiction. Thus, there must be two $x_1, x_2 \in \mathbb{R}$ such that

$$\frac{\sigma(x_1 w_0)}{a_0 x_1 \sigma'(x_1 w_0)} \neq \frac{\sigma(x_2 w_0)}{a_0 x_2 \sigma'(x_2 w_0)}. \tag{29}$$

Therefore, by Lemma 5.2,

$$\nabla G(\cdot, \boldsymbol{\theta}_0)|_{\boldsymbol{\theta}_0^*} = 0. \tag{30}$$

(b) Let $\boldsymbol{x} \neq 0$ and let $\{\boldsymbol{x}/|\boldsymbol{x}|, \boldsymbol{b}_1, ..., \boldsymbol{b}_{M-1}\}$ be an orthonormal basis of $\mathbb{R}^M$. For any parameter $\boldsymbol{\theta} = (w_x \boldsymbol{x}/|\boldsymbol{x}| + \sum_{i=1}^{M-1} w_i \boldsymbol{b}_i, a)$,

$$L(\boldsymbol{\theta}, S) = |a\sigma(\boldsymbol{x}^{\mathrm{T}}\boldsymbol{w}) - y|^2 = |a\sigma(|\boldsymbol{x}|w_x) - y|^2. \tag{31}$$

Therefore by (a), the trajectory of GF for $L(\cdot, S)$ starting at $\boldsymbol{\theta}_0 = (\boldsymbol{w}_0, a_0)$ with $a_0 \neq 0$ ends at a distinct point $\boldsymbol{\theta}_0^* = (\boldsymbol{w}_0^*, -a_0^*)$, and the partial derivative of $G(\cdot, \boldsymbol{\theta}_0)$ with respect to $\boldsymbol{x}/|\boldsymbol{x}|$ and $a$ vanish. By letting $\boldsymbol{x}$ be the multiples of each of the standard basis of $\mathbb{R}^M$, we can see that $\nabla G(\cdot, \boldsymbol{\theta}_0)|_{\boldsymbol{\theta}_0^*} = 0$.

$\square$

*Proof of Theorem 6.2.*

(a) Suppose that $G \in C^1(\mathbb{R}^M \times \mathbb{R}^M)$ is strongly convex. For any $\boldsymbol{\theta}_0$, $\nabla G(\cdot, \boldsymbol{\theta}_0)|_{\boldsymbol{\theta}} = 0$ for at most one $\boldsymbol{\theta} \in \mathbb{R}^M$. Fix $\boldsymbol{w}_0 \in \mathbb{R}^{M-1}$ and $a_0 \in \mathbb{R}\backslash\{0\}$. Since $\sigma, \sigma'$ are strictly positive, Lemma 6.2 says that $\nabla G(\cdot, (\boldsymbol{w}_0, a_0))|_{(\boldsymbol{w}_0, -a_0)} = 0$. By choosing two different values of $a_0$, we can see that there are two points at which $\nabla G(\cdot, \boldsymbol{\theta}_0) = 0$. Thus, $G$ is not strongly convex.

(b) Fix $\boldsymbol{w}_0 \in \mathbb{R}^{M-1}$ and $a_0 \in \mathbb{R}\backslash\{0\}$. By Lemma 6.2, $\nabla G(\cdot, (\boldsymbol{w}_0, a_0))|_{(\boldsymbol{w}_0, -a_0)} = 0$; equivalently, $\nabla G(\cdot, 0)|_{(0, -2a_0)} = 0$ for all $a_0 \neq 0$. Since $\nabla G(\cdot, 0)$ is continuous, $\nabla G(\cdot, 0)|_{(0,0)} = 0$. It follows that for any $\boldsymbol{\theta} \in \mathbb{R}^M$, $G(\cdot, \boldsymbol{\theta})$ must be constant on the set $\{(0, p) : p \in \mathbb{R}\}$.

$\square$

*Proof of Corollary 6.2.*

(a) Due to the structure of $g$ we can apply Lemma 6.1 and 6.2 to each pair $(\boldsymbol{w}_k, a_k)$. The idea is to fix any $\boldsymbol{\theta}_0 = (\boldsymbol{w}_{0k}, a_{0k})_{k=1}^m$ and any $\boldsymbol{x} \in \mathbb{R}^d$ such that the components of $\boldsymbol{x}$, $\boldsymbol{x}_l \neq 0$ for all $1 \leq l \leq d$, and $g(\boldsymbol{\theta}_0, \boldsymbol{x}) \neq 0$ and $\sum_{k=1}^m a_{0k}\sigma'(\boldsymbol{x}^{\mathrm{T}}\boldsymbol{w}_{0k}) \neq 0$. Set $S := (\boldsymbol{x}, -g(\boldsymbol{\theta}_0, \boldsymbol{x}))$. Since $L(\boldsymbol{\theta}_0, S) > 0$, the GF for $L$ starting at $\boldsymbol{\theta}_0$ is not constant.

For each $1 \leq k \leq m$ we have

$$\dot{a}_k(t) = -\frac{\partial L}{\partial a_k}(\boldsymbol{\theta}, S) = -2(g(\boldsymbol{\theta}, \boldsymbol{x}) - y)\sigma(\boldsymbol{x}^{\mathrm{T}}\boldsymbol{w}_k(t))$$

$$\dot{\boldsymbol{w}}_k(t) = -\frac{\partial L}{\partial \boldsymbol{w}_k}(\boldsymbol{\theta}, S) = -2(g(\boldsymbol{\theta}, \boldsymbol{x}) - y)\sigma'(\boldsymbol{x}^{\mathrm{T}}\boldsymbol{w}_k(t))a_k(t)\boldsymbol{x}.$$

This means for any $1 \leq k \leq m$ and any $1 \leq l \leq d$,

$$\dot{a}_k(t)a_k(t) = \frac{(\dot{\boldsymbol{w}}_k)_l(t)\sigma(\boldsymbol{x}^{\mathrm{T}}\boldsymbol{w}_k(t))}{\boldsymbol{x}_l\sigma'(\boldsymbol{x}^{\mathrm{T}}\boldsymbol{w}_k(t))}.$$

By the proof of Lemma 6.2 (see Lemma A.4), there is some strictly monotonic $h$ depending only on $\boldsymbol{x}_l$ and $\sigma$, such that $\frac{1}{2}a_k^2(t) - \frac{1}{2}a_k^2(0) = h(\boldsymbol{w}_k(t)) - h(\boldsymbol{w}_k(0))$. Thus, applying the proof of Lemma 6.2, we can see that $\lim_{t\to\infty} a_k(t) = -a_{0k}$ and $\lim_{t\to\infty} \boldsymbol{w}_k(t) = \boldsymbol{w}_{0k}$ for each $k$. Moreover, since $\boldsymbol{x}_l \neq 0$,

$$\lim_{t\to\infty} \frac{\dot{a}_k(t)}{(\dot{\boldsymbol{w}}_k)_l(t)} = -\frac{g(\boldsymbol{\theta}_0, \boldsymbol{x})}{\boldsymbol{x}_l \sum_{k=1}^m a_{0k}\sigma'(\boldsymbol{w}_{0k}\boldsymbol{x})}.$$

Again, argue in the same way as in Lemma A.4, we conclude that by choosing different $\boldsymbol{x}$, $\partial_{a_j}G(\cdot, (\boldsymbol{w}_{0k}, a_{0k})_{k=1}^m), \partial_{\boldsymbol{w}_j}G(\cdot, (\boldsymbol{w}_{0k}, a_{0k})_{k=1}^m)$ vanish at $(\boldsymbol{w}_{0k}, -a_{0k})_{k=1}^m$, for each $1 \leq j \leq m$.

Now argue in the same way as in Theorem 6.2 by choosing two different initial points. Then the derivative of $G$ vanishes at two points, whence $G$ cannot be strongly convex.

(b) Fix $\boldsymbol{w}_{01}, ..., \boldsymbol{w}_{0m}$ and choose different $a_{01}, ..., a_{0m}$. Since $G \in \mathcal{G}_M$, $\nabla G(\cdot, 0)|_{(0, -2a_{0k})_{k=1}^m} = 0$ for almost all $(a_{01}, ..., a_{0m}) \in \mathbb{R}^m$. Thus, for any $\boldsymbol{\theta} \in \mathbb{R}^{(d+1)m}$, $G(\cdot, \boldsymbol{\theta})$ must be constant on the set $\{(0, p_k)_{k=1}^m : p_k \in \mathbb{R}\}$.

$\square$

**Proposition A.2** (Proposition 6.1). *Let $w \in \mathbb{R}$. Fix a point $(x_0, y_0) \in \mathbb{R}^2$ with $y_0 \neq 0$. Let $F : \mathbb{R}^2 \to \mathbb{R}$, $F(p, x) = \tilde{\sigma}(xw)\tilde{\sigma}(x_0p) - \tilde{\sigma}(xp)\tilde{\sigma}(x_0w)$. We have*

(a) *Suppose that $|F(p, x)| \geq C|p - w|^k|x - x_0|^r$ for some $C > 0$ and $r, k \in \mathbb{N}$ near $(w, x_0)$. Then for sufficiently small $\delta > 0$, if $0 < |x - x_0| < \delta$, $y \neq 0$ and $y\tilde{\sigma}(xw) = y_0\tilde{\sigma}(x_0w)$, there is no $p \in \mathbb{R}$ such that $0 < |p - w| < \delta$ and $y\tilde{\sigma}(xp) = y_0\tilde{\sigma}(x_0p)$.*

(b) *Suppose that $\tilde{\sigma} \in C^2$ and $\tilde{\sigma}(x_0w), \tilde{\sigma}'(x_0w) \neq 0$. Also suppose*

$$\frac{1}{w} - x_0\left[\frac{\tilde{\sigma}'(x_0w)}{\tilde{\sigma}(x_0w)} - \frac{\tilde{\sigma}''(x_0w)}{\tilde{\sigma}'(x_0w)}\right] \neq 0. \tag{32}$$

*Then for sufficiently small $\delta > 0$, if $0 < |x - x_0| < \delta$, $y \neq 0$ and $y\tilde{\sigma}(xw) = y_0\tilde{\sigma}(x_0w)$, there is no $p \in \mathbb{R}$ such that $0 < |p - w| < \delta$ and $y\tilde{\sigma}(xp) = y_0\tilde{\sigma}(x_0p)$. If, however, $DF(p, x_0) \equiv 0$ for $p$ near $w$ or $DF(w, x) \equiv 0$ for $x$ near $x_0$, then $\sigma$ is a power function near $x_0w$, i.e., $\sigma(x) = Cx^d$ for some $C, d \in \mathbb{R}$, when $x$ is sufficiently close to $x_0w$.*

*Proof.*

(a) Let $g(p) = y\tilde{\sigma}(xp)$ and $g_0(p) = y_0\tilde{\sigma}(x_0p)$. Suppose that the values of $g$ and $g_0$ coincide at $w, p$. Then we have

$$y\tilde{\sigma}(xw) = y_0\tilde{\sigma}(x_0w);$$
$$y\tilde{\sigma}(xp) = y_0\tilde{\sigma}(x_0p). \tag{33}$$

Equivalently, whenever $y \neq 0$ and $y_0 \neq 0$,

$$\tilde{\sigma}(xw)\tilde{\sigma}(x_0 p) - \tilde{\sigma}(xp)\tilde{\sigma}(x_0 w) = F(p, x) = 0. \tag{34}$$

Now by hypothesis, there is some $\delta, \varepsilon > 0$ such that for any $(p, s) \in \mathbb{R}^2$ with $\|(p, x) - (w, x_0)\|_\infty < \delta$, $|F(p, x)| \geq C|p - w|^k |x - x_0|^r > 0$. But this is equivalent to saying that for sufficiently small $\delta > 0$, if $0 < |x - x_0| < \delta$, $y \neq 0$ and $y\tilde{\sigma}(xw) = y_0\tilde{\sigma}(x_0 w)$, there is no $p \in \mathbb{R}$ such that $0 < |p - w| < \delta$ and $y\tilde{\sigma}(xp) = y_0\tilde{\sigma}(x_0 p)$.

(b) Fix $w \neq 0$. Note that

$$
\begin{aligned}
F(p, x) &= \tilde{\sigma}(xw)\tilde{\sigma}(x_0 p) - \tilde{\sigma}(xp)\tilde{\sigma}(x_0 w) \\
&= [\tilde{\sigma}(x_0 w) + \tilde{\sigma}'(x_0 w)(x - x_0)w + o(x - x_0)]\tilde{\sigma}(x_0 p) \\
&\quad - [\tilde{\sigma}(x_0 p) + \tilde{\sigma}'(x_0 p)(x - x_0)p + o(x - x_0)]\tilde{\sigma}(x_0 w) \\
&= \tilde{\sigma}'(x_0 w)\tilde{\sigma}(x_0 p)(x - x_0)w - \tilde{\sigma}'(x_0 p)\tilde{\sigma}(x_0 w)(x - x_0)p + o(x - x_0) \cdot (\tilde{\sigma}(x_0 p) - \tilde{\sigma}(x_0 w)) \\
&= (x - x_0)[\tilde{\sigma}'(x_0 w)\tilde{\sigma}(x_0 p)w - \tilde{\sigma}'(x_0 p)\tilde{\sigma}(x_0 w)p] + o((x - x_0)(p - w)).
\end{aligned}
\tag{35}
$$

It suffices to show that $\tilde{\sigma}'(x_0 w)\tilde{\sigma}(x_0 p)w - \tilde{\sigma}'(x_0 p)\tilde{\sigma}(x_0 w)p = \Omega(p - w)$ for all $p$ sufficiently near $w$. Then we can find some $C > 0$ such that $|\tilde{\sigma}'(x_0 w)\tilde{\sigma}(x_0 p)w - \tilde{\sigma}'(x_0 p)\tilde{\sigma}(x_0 w)p| \geq C|p - w|$ and $o((x - x_0)(p - w)) \leq C|x - x_0||p - w|/2$. Then

$$|F(p, x)| \geq |x - x_0||p - w|C - o((x - x_0)(p - w)) \geq \frac{C}{2}|x - x_0||p - w|, \tag{36}$$

when $x$ is sufficiently close to $x_0$ and $p$ sufficiently close to $w$. Thus, by (a) there is some $\delta > 0$ such that for small enough $\delta > 0$, if $0 < |x - x_0| < \delta$, $y \neq 0$ and $y\tilde{\sigma}(xw) = y_0\tilde{\sigma}(x_0 w)$, there is no $p \in \mathbb{R}$ such that $0 < |p - w| < \delta$ and $y\tilde{\sigma}(xp) = y_0\tilde{\sigma}(x_0 p)$. Since $\tilde{\sigma} \in C^2$, when $\tilde{\sigma}'(x_0 w), \tilde{\sigma}''(x_0 w) \neq 0$, this is equivalent to proving

$$\frac{\tilde{\sigma}'(x_0 p)}{\tilde{\sigma}'(x_0 w)} - \frac{w}{p}\frac{\tilde{\sigma}(x_0 p)}{\tilde{\sigma}(x_0 w)} = \Omega(p - w) \tag{37}$$

for all $p$ near $w$. Note that we have

$$
\begin{aligned}
\frac{\tilde{\sigma}'(x_0 p)}{\tilde{\sigma}'(x_0 w)} &= \frac{\tilde{\sigma}'(x_0 w) + \tilde{\sigma}''(x_0 w)x_0(p - w) + o(p - w)}{\tilde{\sigma}'(x_0 w)} \\
&= 1 + \frac{\sigma''(x_0 w)}{\tilde{\sigma}'(x_0 w)}x_0(p - w) + o(p - w)
\end{aligned}
\tag{38}
$$

and

$$
\begin{aligned}
\frac{w}{p}\frac{\tilde{\sigma}(x_0 p)}{\tilde{\sigma}(x_0 w)} &= \left(\frac{w - p}{p} + 1\right)\frac{\tilde{\sigma}(x_0 w) + \tilde{\sigma}(x_0 w)x_0(p - w) + o(p - w)}{\tilde{\sigma}(x_0 w)} \\
&= \left(\frac{w - p}{p} + 1\right)\left(1 + \frac{\tilde{\sigma}'(x_0 w)}{\tilde{\sigma}(x_0 w)}x_0(p - w) + o(p - w)\right) \\
&= 1 + \frac{w - p}{p} + \frac{\tilde{\sigma}'(x_0 w)}{\tilde{\sigma}(x_0 w)}x_0(p - w) + o(p - w).
\end{aligned}
\tag{39}
$$

Thus, the left side of (37) becomes

$$\frac{\tilde{\sigma}'(x_0 p)}{\tilde{\sigma}'(x_0 w)} - \frac{w}{p}\frac{\tilde{\sigma}(x_0 p)}{\tilde{\sigma}(x_0 w)} = \left[\frac{\tilde{\sigma}''(x_0 w)}{\tilde{\sigma}'(x_0 w)}x_0 + \frac{1}{p} - \frac{\tilde{\sigma}'(x_0 w)}{\tilde{\sigma}(x_0 w)}x_0\right](p - w) + o(p - w). \tag{40}$$

By hypothesis, there is some $\delta > 0$ and some $C > 0$ such that for any $p \in (w - \delta, w + \delta)$,

$$\left|\frac{1}{p} - x_0\left[\frac{\tilde{\sigma}'(x_0 w)}{\tilde{\sigma}(x_0 w)} - \frac{\tilde{\sigma}''(x_0 w)}{\tilde{\sigma}'(x_0 w)}\right]\right| \geq C. \tag{41}$$

This proves (37), which in turn completes the first part of our proof.

Now assume that $DF(w, x) \equiv 0$ for $x$ near $x_0$. Thus, there is a neighborhood $U$ of $x_0$ on which $\frac{\partial F}{\partial p}(w, x) = x_0\tilde{\sigma}(xw)\tilde{\sigma}'(x_0w) - x\tilde{\sigma}'(xw)\tilde{\sigma}(x_0w)$ vanishes. Because $x_0 \neq 0$ and by hypothesis, $\sigma'(x_0w) \neq 0$, we can make this $U$ so small that $0 \notin U$ and $\sigma'(xw) \neq 0$ for $x \in U$. This means

$$\frac{\tilde{\sigma}'(xw)}{\tilde{\sigma}(xw)} = \frac{x_0\tilde{\sigma}'(x_0w)}{\tilde{\sigma}(x_0w)}\frac{1}{x}. \tag{42}$$

Arguing in the same way as the proof of Lemma 6.2, we can see that there are non-zero constants $A, B$ such that

$$\log(A\tilde{\sigma}(xw)) = \frac{x_0\tilde{\sigma}'(x_0w)}{\tilde{\sigma}(x_0w)}\log(Bx). \tag{43}$$

Therefore $\tilde{\sigma}$, and thus $\sigma$, is a power function. Similarly, when $DF(p, x_0) = 0$ for $p$ near $w$, we can integrate and deduce that $\sigma$ is a power function near $w$.

$\square$

**Corollary A.2** (Corollary 6.1). *Following the notations in Proposition 6.1, all the results below hold.*

(a) *Any $\sigma$ and $w, x_0 \neq 0$ such that $\sigma$ is a power function on a neighborhood of $x_0w$ (this includes ReLU and PReLU and Heaviside) satisfies $F = 0$ near $(w, x_0)$.*

(b) *For any analytic activation $\sigma$ and any $x_0, w \in \mathbb{R}$ such that the zero locus of the function $F(p, x) = \tilde{\sigma}(xw)\tilde{\sigma}(x_0p) - \tilde{\sigma}(xp)\tilde{\sigma}(x_0w)$ satisfies*

$$F^{-1}\{0\} \cap U = \{(p, x) \in U : p = w\} \cup \{(p, x) \in U : x = x_0\}$$

*for some neighborhood $U \subseteq \mathbb{R}^2$ of $(w, x_0)$, we can find a sufficiently small $\delta > 0$ such that if $0 < |x - x_0| < \delta$, $y \neq 0$ and $y\tilde{\sigma}(xw) = y_0\tilde{\sigma}(x_0w)$, there is no $p \in \mathbb{R}$ with $0 < |p - w| < \delta$ and $y\tilde{\sigma}(xp) = y_0\tilde{\sigma}(x_0p)$.*

(c) *If $\sigma = e^x$ or $\sigma = e^{-x^2}$, then for any $x_0 \in \mathbb{R}$, we can find a sufficiently small $\delta > 0$ such that if $0 < |x - x_0| < \delta$, $y \neq 0$ and $y\tilde{\sigma}(xw) = y_0\tilde{\sigma}(x_0w)$, there is no $p \in \mathbb{R}$ with $0 < |p - w| < \delta$ and $y\tilde{\sigma}(xp) = y_0\tilde{\sigma}(x_0p)$.*

(d) *Let $w > 0$. If $\sigma = \frac{1}{1+e^{-x}}$, for any $x_0 \in (-\infty, w^{-1}) \cup (2w^{-1}, \infty)$, we can find a sufficiently small $\delta > 0$ such that if $0 < |x - x_0| < \delta$, $y \neq 0$ and $y\tilde{\sigma}(xw) = y_0\tilde{\sigma}(x_0w)$, there is no $p \in \mathbb{R}$ with $0 < |p - w| < \delta$ and $y\tilde{\sigma}(xp) = y_0\tilde{\sigma}(x_0p)$.*

*Proof.*

(a) Note that $\sigma$ is a power function near $w$ if and only if $\tilde{\sigma}$ is. Therefore, suppose that $\tilde{\sigma}(x) = x^q$ for some $q \in \mathbb{R}$, then for any $(x, p)$ sufficiently close to $(x_0, w)$,

$$F(p, x) = (xw)^q(x_0p)^q - (xp)^q(x_0w)^q = 0.$$

(b) Apply Lojasiewicz distance inequality to $F^2$ on $U$. Denote $\ell_1 := \{(p, x) \in \mathbb{R}^2 : p = w\}$ and $\ell_2 := \{(p, x) \in \mathbb{R}^2 : x = x_0\}$. Since $(F^2)^{-1}\{0\} = F^{-1}\{0\}$, there are some $C, \beta > 0$ such that for $(p, x) \in U$ sufficiently close to $(w, x_0)$, we have

$$\begin{aligned}
|F^2(p, x)| &\geq C\mathrm{dist}\left((p, x), F|_U^{-1}\{0\}\right)^\beta \\
&= C\mathrm{dist}\left((p, x), F^{-1}\{0\} \cap U\right)^\beta \\
&\geq C\min\{\mathrm{dist}\left((p, x), \ell_1\right)^\beta, \mathrm{dist}\left((p, x), \ell_2\right)^\beta\} \\
&= C\min\{|x - x_0|^\beta, |p - w|^\beta\} \\
&\geq C|x - x_0|^\beta|p - w|^\beta.
\end{aligned}$$

This shows that the assumption in Proposition 6.1 (a) is satisfied, so the desired result follows.

(c) Note that both $e^x$ and $e^{-x^2}$ are (real) analytic functions. We will prove by applying the result in (b). For $\sigma(x) = e^x$, when $F(p, x) = 0$ we must have

$$e^{-xw}e^{-x_0 p} = e^{-xp}e^{-x_0 w},$$

which gives $xw + x_0 p = xp + x_0 w$, i.e., $(x - x_0)(p - w) = 0$. Thus, either $p = w$ or $x = x_0$. Similarly, for $\sigma(x) = e^{-x^2}$, when $F(p, x) = 0$ we must have

$$e^{x^2 w^2}e^{x_0^2 p^2} = e^{x^2 p^2 + x_0^2 w^2}.$$

Thus, $x^2 w^2 + x_0^2 p^2 = x^2 p^2 + x_0^2 w^2$, which gives $(x^2 - x_0^2)(p^2 - w^2) = 0$ and this holds when $x^2 = x_0^2$ or $w^2 = p^2$. Thus, for $(p, x)$ sufficiently close to $(w, x_0)$, we must have $x = x_0$ or $p = w$.

The proof above shows that either for $\sigma(x) = e^x$ or $\sigma(x) = e^{-x^2}$, we can find some neighborhood $U \subseteq \mathbb{R}^2$ of $(w, x_0)$ such that

$$F^{-1}\{0\} \cap U = \{(p, x) \in U : p = w\} \cup \{(p, x) \in U : x = x_0\}.$$

Therefore, the desired result follows from (b).

(d) $\tilde{\sigma}(x) = 1 + e^{-x}$. Let $u = x_0 w$. Therefore,

$$1 - u\left(\frac{\tilde{\sigma}'(u)}{\tilde{\sigma}(u)} - \frac{\tilde{\sigma}''(u)}{\tilde{\sigma}'(u)}\right) = 1 - \frac{x}{1 + e^{-x}}. \tag{44}$$

Let $z$ denote the root of $1 - \frac{x}{1+e^{-x}}$. Since $z \in (1, 2)$, it follows that when $u \in (-\infty, 1) \cup (2, \infty)$, the desired result follows from Proposition 6.1.

$\square$

