# OpenReview forum: "Limitation of Characterizing Implicit Regularization by Data-independent Functions"
_TMLR — Accepted by TMLR_

### Review · Reviewer_ZwaV · 2023-07-17

**Summary Of Contributions:**

This paper provides a thorough investigation on the notion of implicit regularization in neural networks trained using gradient descent. The authors propose two dynamic mechanisms, namely the Two-point Overlapping Mechanism and the One-point Overlapping Mechanism, which they claim can impose strict limitations on the function that characterizes implicit regularization, or potentially render such characterization infeasible. The authors also present specific instances of neural networks with a single hidden neuron and widely used activation functions that can demonstrate each of these mechanisms. Additionally, the authors provide a collection of guidelines for generating diverse categories of one-hidden-neuron neural networks capable of implementing these mechanisms.


**Audience:**

Yes

**Broader Impact Concerns:**

None.

**Claims And Evidence:**

Yes

**Requested Changes:**

1. Provide a discussion on 1) how can we genralize the results to neurla networks with more than one hidden neuron; 2) how can we extend the results to neural networks with ReLU activation.

2. Discuss the practical implications of their findings in more detail.

**Strengths And Weaknesses:**

# Strengths

1. The paper is well-structured and provides a clear and detailed explanation of the concepts. The authors' approach to characterizing implicit regularization using data-independent functions is novel and provides a new perspective on understanding the behavior of deep learning models. The proposed overlapping mechanisms and the corresponding examples are insightful and help to illustrate the authors' arguments effectively.

2. The authors provide a mathematical definition and study of implicit regularization, which is a central issue in deep learning theory. This contrast the prior literatures, where the implict regualrization is mostly not well-defined or not rigorous.

3. The authors propose two dynamical mechanisms and provide two recipes for creating classes of one-hidden-neuron NNs that cannot be fully characterized by a type of or all data-independent functions.

4. The paper makes a contribution by highlighting the profound data dependency of implicit regularization in general. It may inspire future studies on the data dependency of NN implicit regularization.

# Weakness

1. The constructions only work for neural networks with only one hidden neuron and it's unclear how can it  generalize to the general neural networks that are closer to the one used in pratice. Thus the results in the paper may not fully represent the complexity and diversity of real-world neural networks.

2. Additionally, the theoretical results only work for continuously differentiable activation function such as sigmoid and softplus, and it's unclear how can we generalize the results to neural networks with activation functions like ReLU.

3. It's not so surprising that the implicit regularization may dependent on the data, especially given the observation that neurla networks can fit the data with random labels perfectly (e.g., Zhang et al., 2017). Therefore, it's unclear to me how the results in the paper can foster our understanding on the implict regualrization even after read this paper.

---

> ### Author Response · Authors · 2023-08-09
>
> We really appreciate your thoughtful and constructive comments. We have made the following changes and explanation.
>
> Requested Changes
>
> Point 1.
>
> How can we generalize the results to neural networks with more than one hidden neuron?
>
> Reply:
>
> See Response to General points.
>
> Point 2.
>
> How can we extend the results to neural networks with ReLU activation?
>
> Reply:
>
> Vardi \& Shamir (2021) has already shown that single neuron ReLU network has data-dependent implicit regularization. Also, we note that the loss landscape for a ReLU network is basically piecewise linear, so the construction of data-dependency of its implicit regularization relies on the non-linearity at the connection of pieces. This is different from our two recipes.
>
> Point 3.
>
> Discuss the practical implications of their findings in more detail.
>
> Reply:
>
> Following Vardi \& Shamir (2021), our work provides further evidence about the data dependency of the implicit regularization underlying the training of the single neuron network. Our results mainly contribute to the theoretical understanding of the implicit regularization of DNNs. Regarding the practical implication, we add the following comment in Section 7.2 in Conclusion and Discussion in the revised manuscript:
>
> "..., whether a data-dependent implicit regularization could help the generalization of DNNs remains an open problem for the future research."
>
> Other Points
>
> Point 4.
>
> It’s not so surprising that the implicit regularization may depend on the data, especially given the observation that neural networks can fit the data with random labels perfectly. Therefore, it’s unclear to me how the results in the paper can foster our understanding on the implicit regularization even after read this paper.
>
> Reply:
>
> That neural networks can fit data with random labels perfectly does not imply the implicit regularization is data-dependent. For example, a large deep neural network at the NTK regime can also fit random labels perfectly. However, it is well studied that its implicit regularization can be characterized by the l2 norm in the parameter space independent to data. In fact, Zhang et.al. (2017) demonstrates the existence of implicit regularization, but not how the implicit regularization can be characterized. Our paper dives into this issue and provides further evidence that many commonly considered data-independent functions like various norms in the parameter space may not fully characterize the implicit regularization of NNs. Our results suggest focusing on the data dependent aspect of implicit regularization in the future studies.

---

### Review · Reviewer_RN8d · 2023-07-21

**Summary Of Contributions:**

This work studies implicit regularization of gradient flow learning in NN models.  The authors

1. provided formal definitions for implicit and explicit regularization,
2. proposed two mechanisms that make any data-independent characterization for implicit regularization impossible, and a recipe to construct examples that realize one or both of the mechanisms,
3. showed that across a wider range of single-hidden-layer NN models, the implicit regularization effect cannot be precisely characterized by data-independent regularization.

The results improved upon a previous work (Vardi & Sharmi, 2021) that studied ReLU NNs, primarily by allowing for more general activation functions.

**Audience:**

Yes

**Claims And Evidence:**

No

**Requested Changes:**

Most importantly, it would be helpful to provide further discussion that justifies the contribution claims, e.g., how the present results may inspire the study of the data dependency of implicit regularization (in a way that past works do not).

There are also a few minor issues:

- Definition 4.2, 4.3: you may want to clarify if $\mathcal{A}$ continues to be defined as above Defn. 4.2, or switch to a different notation to avoid confusion.
- Example (b) below Defn 4.3: $G$ is not formally defined.
- You may want to clarify if $\mathcal{M}_S$ can contain local minima, or if the limit of any GF is always assumed to be a global minima; otherwise statements like Eq. 3 cannot be valid.


**Strengths And Weaknesses:**

**Strengths.**

- The general problem of understanding implicit regularization is important.
- The manuscript is for the most part well-written and provides good intuition for the authors' construction.
- The findings are validated by numerical experiments.

**Weaknesses.**

- My main concern is about the claims on the contributions of this work.  The authors stated that their results "signify the profound data dependency of implicit regularization in general", but this broad message can already be found in Vardi & Sharmi (2021), and the improvements in the main theorems of this work (relaxing the requirements on the activation) appear somewhat technical.

- While the manuscript contains additional, intermediate results (contribution 2 above) that may be of independent interest, the construction appears to be fairly restrictive (e.g., the resulted training set always contains a single sample), so it is unclear if the constructions are indeed more broadly useful, beyond for the proof of the present results.

- There are also some clarity issues which are easily fixable; see below for details.

---

> ### Author Response · Authors · 2023-08-09
>
> We really appreciate your thoughtful and constructive comments. We have made the following changes and explanation.
>
> Requested Changes
>
> Point 1.
>
> Most importantly, it would be helpful to provide further discussion that justifies the contribution claims, e.g., how the present results may inspire the study of the data dependency of implicit regularization (in a way that past works do not).
>
> Reply:
>
> To the best of our knowledge, current study only shows the data dependence results for one-hidden-neuron NNs with specific activations. We take a step further by showing these hold for general two-layer NNs (Corollary 6.2 in revised manuscript) and by demonstrating the mechanism behind it. In the revised manuscript, we also add discussion about the idea to generalize both recipes to multi-layer NNs and multi-sample loss functions. Please see our Response to General Points as well.
>
> Point 2.
>
> Minor issue:
>
> (i) Definition 4.2, 4.3: you may want to clarify if $\mathcal{A}$ continues to be defined as above Defn. 4.2, or switch to a different notation to avoid confusion.
>
> (ii) Example (b) below Defn 4.3: $G$ is not formally defined.
>
> (iii) You may want to clarify if $\mathcal{M}_S$ can contain local minima, or if the limit of any GF is always assumed to be a global minima; otherwise statements like Eq.3 cannot be valid.
>
> Reply:
>
> (i) We clarify the definition of $\mathcal{A}$ in Section 4.1 in the revised manuscript as the set of $A_{\text{min}}$ where $A_{\text{min}}$ finds the global minima of $L_S$.
>
> (ii) We make the following clarification in Section 4.1 in the revised manuscript.
>
> "...The construction of $G$ is possible in a trivial way: we may find some $c, c' \in \mathbb{R}$ with $c,c' > 0$, then set $G(\theta_0^*, \theta_0) = c$ for any $\theta_0 \in \mathbb{R}^M$ and any $\theta_0^* \in A_{GF,\theta_0}$, and $G(\theta^*, \theta) = c'$ otherwise. In certain situation, we can make $G$ behave much better. For example, if the gradient flows are on the loss landscape of a linear regression problem, we may simply set $G(\theta^*, \theta) = | \theta^* - \theta |$ for all $(\theta^*, \theta) \in \mathbb{R}^M \times \mathbb{R}^M$."
>
> (iii) We clarify the definition of $\mathcal{M}_S$ in Section 3 in the revised manuscript as the global minima of $L_S$. $\mathcal{M}_S$ is always assumed so throughout the manuscript.
>
> Other Points
>
> Point 3.
>
> The contributions of this work. The authors stated that their results "signify the profound data dependency of implicit regularization in general'', but this broad message can already be found in Vardi \& Shamir, and the improvements in the main theorems of this work (relaxing the requirements on the activation) appear somewhat technical.
>
> Reply:
>
> Following the reviewer's comment, we have changed the corresponding parts of abstract and the contribution as follows:
>
> (Abstract) "...Following the previous works, our results further emphasize the profound data dependency of implicit regularization in general, inspiring us to study in detail the data dependency of NN implicit regularization in the future."
>
> (Contribution) "...Based on Vardi \& Shamir (2021), we further emphasize the importance of data-dependence of implicit regularization in general, which should be carefully studied for NNs in the future."

---

### Review · Reviewer_qUML · 2023-07-25

**Summary Of Contributions:**

This paper studies the limitation of characterizing implicit regularization by data independent functions. How does overparameterized neural networks find solutions that generalize well has been a central topic in deep learning theory. One common perspective is to argue via the implicit regularization of the underlying training algorithm (e.g. SGD). This paper contribute to this topic by showing that there are many implicit regularizations that cannot be described as data independent functions.

**Audience:**

Yes

**Claims And Evidence:**

Yes

**Requested Changes:**

See above

**Strengths And Weaknesses:**

Strength

- This paper provided formal definition of explicit regularization and implicit regularization (for a gradient flow), and ways to characterize them with data independent functions.
- It provided two mechanism and associated recipes that allows construction of examples with non-linear neural networks to demonstrate that there are implicit regularizations that are impossible to be fully characterized with data independent function.

Weakness

- The observation that there exist implicit regularizations that cannot be fully characterized by data independent function is important. However, the existence of such examples does not necessarily imply that all (or many) of the implicit regularization of neural networks that are trained in real world data fall in the same category as well. It would be great to have some more discussions regarding this.

---

> ### Author Response · Authors · 2023-08-09
>
> We really appreciate your thoughtful and constructive comments. We have made the following changes and explanation.
>
> Weakness and Requested Changes
>
> Point 1.
>
> The observation that there exist implicit regularizations that cannot be fully characterized by data independent function is important. However, the existence of such examples does not necessarily imply that all (or many) of the implicit regularization of neural networks that are trained in real world data fall in the same category as well. It would be great to have some more discussions regarding this.
>
> Reply:
>
> If the GF for a loss function is independent of data, then we can find an explicit regularization of it, following Examples. (b) after Definition 4.3, even if the function could have very complicated behavior. However, in general, it is not clear whether all of the implicit regularization of neural network can be characterized by (one of) our two recipes. We explicitly point this limitation in the revised manuscript in Section 7.2 in Conclusion and Discussion as follows:
>
> "..., it is still not clear whether all the implicit regularization of NNs fall into one of our recipes or mechanisms."

---

### Review · Reviewer_rC1y · 2023-08-04

**Summary Of Contributions:**

The paper propose a study of implicit regularization of neural network. More precisely, the authors question the data dependence of this type of regularization. Their main contributions are the following:
* A clear mathematical definition of regularization, that encompasses both implicit and explicit regularization.
* Building on these definitions, two dynamical mechanisms and derived "Recipes" to produce rich classes of 1-hidden-layer networks for which it is impossible to fully characterize implicit regularization with data-independent functions.

This works builds on previous studies (the most closely related being Vardi & Shamir 2021) and generalize their findings to a broader class of 1-hidden-layer networks.

**Audience:**

Yes

**Claims And Evidence:**

Yes

**Requested Changes:**

Building on the previous rubric:
* Can the author provide examples for implicit regularization, similarly to what they proposed for the explicit category?
* Can the authors comment on or precise the definition on the set of minima they're study is working with?
* (Less important and slightly beyond the scope of this work) It would be really impactful if the authors can provide some comments or insights on how to generalize their study to multi-layer networks.

**Strengths And Weaknesses:**

Strengths:
* The paper tackles an important problem of relevance to the community and can advance our understanding of neural networks.
* Despite being closely related to previous work, I found the paper brings sufficient novelty and new insights.
* The formal definitions that the reasoning is built on make sense.
* The paper is clearly written and well structured. It is easy to read, and I found the simulation particularly clarifying.

Weaknesses:
* One part that is unclear to me is the definition of the set $\mathcal{M}_S$. The way the authors describe it suggests that is contains the global minima of the loss on a dataset $S$.  Further, the described mechanisms suggest that implicit regularization selects a minima from this set. However, there is no obvious reason for me why implicit regularization shouldn't bias the loss and therefore lead to a different set of minima (e.g. see "ON THE ORIGIN OF IMPLICIT REGULARIZATION IN STOCHASTIC GRADIENT DESCENT" , S. L. Smith et al. 2021).
* Related to this, the paper could be make clearer by some practical examples on implicit regularization (settings where in plays a role, some insights on its origins, ...). This can help clarify and test the validity of the definition that the authors propose.

---

> ### Author Response · Authors · 2023-08-09
>
> We really appreciate your thoughtful and constructive comments. We have made the following changes and explanation.
>
> Requested Changes
>
> Point 1.
>
> Can the author provide examples for implicit regularization, similarly to what they proposed for the explicit category?
>
> Reply:
>
> We add the following example in Section 4.1 in the revised manuscript:
>
> "...Consider the gradient flows on the loss landscape of a linear model, i.e., $\{ A_{\text{GF}, \theta_0}: \theta_0 \in \mathbb{R}^M\}$ is the collection of gradient flows with respect to the loss function
>
> $L(\theta, S) = \sum_{i=1}^n |\theta \cdot x_i - y_i|^2$, $\theta \in \mathbb{R}^M$, $n < M$.
>
> For the samples $(x_i, y_i)$ we require that $(x_1, ..., x_n)$ has full rank. Let $\mathcal{A}$ be the set of $A_{\theta_0}$'s, $\theta_0 \in \mathbb{R}^{M}$ where each $A_{\theta_0}$ finds the point in $L^{-1}(0)$ which has the shortest distance to $\theta_0$. Then we obtain a map $\mathcal{R}$ from $\mathcal{A}$ to the set of $A_{\text{GF}, \theta_0}$'s by $\mathcal{R}(A_{\theta_0}) = A_{\text{GF},\theta_0}$."
>
> Point 2.
>
> Can the authors comment on or precise the definition on the set of minima their study is working with?
>
> Reply:
>
> We always work with the set of global minima of the loss function $L_S$. We have precised the meaning of $\mathcal{M}_S$ as global minima in the work as well in Section 3 as follows:
>
> "... If $L_S$ has a minimum, we further denote the set of its global minima by $\mathcal{M}_S$. ''
>
> Point 3.
>
> It would be really impactful if the authors can provide some comments or insights on how to generalize their study to multi-layer networks.
>
> Reply:
>
> Please see Response to General points. We also add a corollary in Section 6 in revised manuscript to show that One-point Overlapping Recipe works for two-layer NNs with multiple neurons:
>
> "Fix $m, d \in \mathbb{N}$. Consider the two-layer neural network $g(\theta, x) = \sum_{k=1}^m a_k \sigma(w_k\cdot x)$, where $\theta = (w_k, a_k)_{k=1}^m \in \mathbb{R}^{(d+1)m}$, and the corresponding loss function
>
> $L(\theta, (x, y)) = |g(\theta, x) - y|^2 = \left| \sum_{k=1}^m a_k \sigma(w_k\cdot x) - y \right|^2$.
>
> Suppose that $\sigma: \mathbb{R} \to \mathbb{R}^+$ is differentiable and strictly increasing. Based on One-point Overlapping Recipe, we have
>
> (a) $L$ cannot be characterized by any strongly convex data-independent function $G \in C^1(\mathbb{R}^{(d+1)m} \times \mathbb{R}^{(d+1)m})$.
>
> (b) If the implicit regularization for $L$ is characterized by a data-independent function $G \in \mathcal{G}_M$ in the weak sense, then for any $\theta \in \mathbb{R}^{(d+1)m}$, $G(\cdot, \theta)$  is constant on an affine subspace of $\mathbb{R}^{(d+1)m}$. "
>
> [Due to limitations of OpenReview coding, we changed some sentences and math formulas. Please refer to our revised manuscript when necessary. Thanks!]

---

### Author Response · Authors · 2023-08-09
**Response to General Points**

Point.

Generalization of our results to cases with more complicated neural networks and multiple samples.

Reply:

We provide further theoretical discussion regarding the generalization of overlapping recipes in a new subsection in Conclusions and Discussion (Subsection 7.1 in the revised manuscript) as follows:

"In this part we briefly discuss the generalization of our One-point Overlapping and Two-point Overlapping recipes (as well as corresponding mechanisms). We discuss the potential for our recipes to work for two-layer (fully-connected) NNs with multiple neurons with one-sample dataset, or even for more general models and loss functions.

Let's start with the Two-point Overlapping Recipe. Indeed, for this recipe, very few restrictions are put on the structure of the network or the loss functions; so in particular it can be generalized to a much larger set of models. To see this, consider a $\sigma$-dependent model $g = g_\sigma: \mathbb{R}^M \times \mathbb{R}^d \to \mathbb{R}$, and $L(\theta, (x, y)) = L_\sigma(\theta, (x, y)) = |g_\sigma(\theta, x) - y|^2$. The key of Two-point Overlapping Recipe is to "construct'' the model $g$ by "constructing'' $\sigma$, meanwhile taking the advantage that a convergent GF uses only partial information of $\sigma$. By looking at this recipe for one-neuron models (Section 6.1 and/or 6.2), to make the Two-point Overlapping Recipe work for $g_\sigma$ we basically need to

(a) Find $\theta_1^*$, some dataset $S_1$ and some $\sigma_1$ such that the GF for $L_{\sigma_1}(\cdot, S_1)$ starting at $\theta_0$ converges to $\theta_1^*$, and $L_{\sigma_1} (\theta_1^*, S_1) = 0$.

(b) Find $\theta_2^*$ such that $g_{\sigma_1}(\theta_2^*, S_1) = y$, namely, $L_{\sigma_1}(\theta_2^*, S_1) = 0$.

(c) Find another dataset $S_2$ and $\sigma_2$ so that $g_{\sigma_2}(\theta_2^*, S_2) = y$, and the GF for $L_{\sigma_2}(\cdot, S_2)$ converges to $\theta_2^*$.

(d) Finally define $\sigma$ by appropriately "concatenating'' $\sigma_1$ and $\sigma_2$.

Note that here we do not require that $S_1, S_2$ must be singletons. As long as the system is over-parametrized, these requirements are easy to satisfy, not only because they set few restrictions on the choice the activations, the samples, and the parameters we choose, but also because the requirements are loosely related to each other, e.g., requirement (b) does not have much to do with requirement (a).

The One-point Overlapping Recipe deals with the relationship between the partial derivatives of the loss function, so it naturally depends more on the structure of both the model and the loss function. We have shown that this recipe works for two-layer fully connected NNs as well. Unfortunately, currently we do not know how to generalize it to NNs with more layers, and/or to multi-sample loss functions. What we know is: to make it work we basically need to

(a) Find two distinct points point $\theta_0, \theta_0^* \in \mathbb{R}^M$ and some datasets $S_1, ..., S_M$.

(b) For each $1 \le j \le M$, the GF $\gamma_j$ for $L_{S_j}$ starting at $\theta_0$ converges to $\theta_0^*$.

(c) For each $1 \le j \le M$, $\gamma_j$ has a limiting direction, i.e., $\lim_{t\to\infty} \frac{\gamma_j(t)}{|\gamma_j(t)|}$ exists; moreover, these directions are linearly independent.

With such information we can conclude that $\nabla G(\cdot, \theta_0)|_{\theta_0^*} = 0$ as in Lemma 6.2."

We also add the following discussion in Section 7.2 in Conclusions and Discussion:

"...Furthermore, as we have discussed before, our recipes and mechanisms have the potential to be extended to two-layer NNs with multiple neurons, or even more general models. In comparison, the existing examples mainly focus on more specific cases (e.g., specific set-up or specific kind of activations). The generality of our recipes thus suggests that it is generally difficult to characterize implicit regularization by data-independent functions, if not impossible.

On the other hand, our work does not fully explain the implicit regularization in NNs. For example, we do not know whether all the implicit regularization of NNs fall into one of our recipes and/or mechanisms. Neither are we clear about the practical implication of it. In particular, whether a data-dependent implicit regularization could help the generalization of NNs remains an open problem for the future research...''

[Due to limitations of OpenReview coding, we changed some math formulas. Please refer to our revised manuscript when necessary. Thanks!]

---

### Decision · Action_Editor_Rr2n · 2023-10-26

**Recommendation:** Accept with minor revision

**Comment:**

Reviewers generally agreed that the manuscript deals with a timely and important topic, is well-written, and delivers novel contributions.  Several comments were raised regarding relevance to practical neural networks, significance over past work (in particular Vardi & Shamir 2021) and certain technical details.  These comments were largely addressed, and in my opinion the paper in its current form is almost ready for publication.  The only point I would further clarify is that by reviewer rC1y about the set M_S comprising global minima.  From my experience in the area assuming convergence to global minima in studying implicit regularization is quite common; I recommend the authors add a short emphasis of this.  Overall the paper is a worthy addition to TMLR.

**Audience:**

The topic of the manuscript is of high interest to TMLR readership

**Claims And Evidence:**

Claims made in the manuscript are well supported